# Guided Structural Inference: Leveraging Priors with Soft Gating Mechanisms

Aoran Wang [1]   Xinnan Dai [2]   Jun Pang [1][3]

## Abstract

Existing methods for inferring latent relational structures struggle to integrate partial prior knowledge, such as known edges, node-degree constraints, and global sparsity, without destabilizing training or conflicting with probabilistic assumptions. We propose Soft-Gated Structural Inference (SGSI), a VAE framework that seamlessly incorporates domain constraints via (1) soft gating with learnable edge masks to preserve gradients, (2) cloning-clamping of deterministic edges to avoid distributional conflicts, and (3) adaptive regularization to balance data-driven learning with domain constraints. By excluding known edges from stochastic inference, SGSI reallocates capacity to uncertain interactions, optimizing the information bottleneck trade-off. Experiments on 16 datasets show SGSI improves edge recovery by up to $9\%$ AUROC over baselines, scales to larger graphs ($94.2\%$ AUROC), and maintains stable training. SGSI bridges domain expertise with data-driven learning, enabling interpretable and robust structural discovery in dynamical systems.

## 1. Introduction

The ability to infer latent relational structures, such as interactions between particles in physics (Kwapień & Drożdż, 2012; Ha & Jeong, 2021; Wu et al., 2024), agents in robotics (Brasó & Leal-Taixé, 2020; Li et al., 2022), or genes in biological systems (Tsubaki et al., 2019; Pratapa et al., 2020), is critical for modeling complex dynamical systems (Birkhoff, 1927; Katok & Hasselblatt, 1995). While recent advances in variational autoencoder (VAE)-based methods (Kipf et al., 2018; Alet et al., 2019; Chen et al., 2021; Löwe et al., 2022; Wang & Pang, 2022; Wang et al., 2023; Wang & Pang, 2024b) have enabled end-to-end learning and inference of these structures, they often overlook a key practical reality: domain experts frequently possess partial prior knowledge on the structure, such as known interactions in biochemical networks (Pratapa et al., 2020) or kinematic constraints in multi-agent systems (Brasó & Leal-Taixé, 2020). These kinds of prior knowledge are ignored, wasting opportunities to guide inference and forcing models to "re-discover" known edges, wasting capacity on irrelevant uncertainty. However, integrating such knowledge into structural inference frameworks is challenging, as naively overwriting edges or enforcing rigid constraints risks destabilizing training, contradicting the probabilistic assumptions of VAEs, and conflicting with the information-theoretic principles underlying these models.

Existing approaches struggle to reconcile prior knowledge with data-driven learning. For instance, forcibly clamping edges to fixed values disrupts gradient flow during backpropagation, while imposing exact sparsity or node-degree constraints creates conflicts with the variational information bottleneck (VIB) objective (Alemi et al., 2017), which balances compression of irrelevant information against predictive accuracy. These issues often manifest as failed convergence or suboptimal inference, limiting the utility of structural inference in practice.

We address these challenges with **Soft-Gated Structural Inference (SGSI)**, a framework that integrates partial prior knowledge through three principal components. First, a soft gating mechanism, where each edge is assigned a learnable parameter $\alpha_e \in [0, 1]$, smoothly masks interactions while preserving gradient flow. Second, known edges are cloned and clamped to deterministic values (1 for present, 0 for absent) during forward passes, avoiding in-place modifications and excluding them from the Kullback-Leibler (KL) divergence calculation. Third, global sparsity and node-degree constraints are enforced via adaptive penalties, allowing deviations when supported by data. By treating known edges as deterministic (zero entropy) and uncertain edges as stochastic, SGSI aligns with the VIB principle: it compresses irrelevant interactions while preserving predictive capacity for uncertain ones, optimizing the trade-off between domain priors and data-driven discovery.

Experiments across dynamical systems and biological networks demonstrate SGSI's advantages. When partial prior

---

[1]Department of Computer Science, University of Luxembourg [2]ShanghaiTech University [3]Institute for Advanced Studies, University of Luxembourg. Correspondence to: Jun Pang <jun.pang@uni.lu>.

*Proceedings of the 42nd International Conference on Machine Learning*, Vancouver, Canada. PMLR 267, 2025. Copyright 2025 by the author(s).

knowledge is available, SGSI improves edge recovery by up to $9\%$ in $\Delta$AUROC score compared to VAE baselines. It enables smooth scaling to larger graphs with $94.2\%$ AUROC, and it maintains stable training where naive implementations diverge. Theoretically, SGSI's cloning-and-clamping strategy "frees" information-theoretic capacity for uncertain edges, avoiding distributional conflicts that plague naive approaches. Practically, inferred structures align with domain expectations, enhancing interpretability without sacrificing predictive accuracy. Our contributions are threefold:

- A **novel approach** with gating mechanisms, clamp-cloning operation and adaptive regularization that integrates prior knowledge (known edges, sparsity, node degrees) into VAEs without gradient instability or distributional conflicts.

- A **theoretical framework** linking SGSI to the VIB principle, showing how deterministic and stochastic edges optimize the compression-prediction trade-off.

- **Empirical validation** across physical, biological, and multi-agent systems, showcasing the performance of SGSI comparing with baseline methods.

## 2. Related Work

**VAE-based Structural Inference.** Many works use variational autoencoders to learn latent graph structures for interacting systems (Kipf et al., 2018; Alet et al., 2019; Chen et al., 2021; Wang & Pang, 2022; Wang et al., 2023). For example, NRI (Kipf et al., 2018) applies a GNN-based decoder to predict future states given a learned adjacency. Key advancements include factorizing interactions for multi-relational systems systems (Webb et al., 2019), integrating efficient message-passing (Chen et al., 2021), incorporating modular meta-learning (Alet et al., 2019), iteratively pruning indirect connections (Wang & Pang, 2022), developing structural inference with reservoir computing (Wang et al., 2023), and estimating partial correlation coefficients based on node embeddings (Wang & Pang, 2024b). Yet, they lack systematic support for known-absent edges, node-degree limits, or global sparsity, and often rely on brittle adjacency overwrites that disrupt training dynamics.

**Partial-Knowledge Approaches.** Some frameworks can handle *some* prior knowledge, e.g., ALaSI (Wang & Pang, 2023) integrates known-present edges via active learning, or SICSM (Wang & Pang, 2024a) can fix certain present edges. However, they lack flexible ways to handle absent edges, node-degree constraints, or global sparsity. Others rely on naive adjacency overwrites that can break gradient flow or create contradictory KL signals.

We address these challenges by introducing *soft gating*, a mechanism that systematically accommodates known edges, global sparsity, and node-degree constraints *without* any in-place modifications or distributional conflicts.

## 3. Preliminaries

### 3.1. Notation

We consider $N$ nodes with observed states $\{x_i(t)|i = 1,\ldots,N; \ t = 1,\ldots,T\}$ over discrete time steps $t \in \{1,\ldots,T\}$, and each $x_i(t) \in \mathbb{R}^d$. Moreover, we combine all the time steps into a single trajectory for node $i$: $X_i = [x_i(1), x_i(2), \ldots, x_i(T)] \in \mathbb{R}^{d\times T}$. We write $\mathbf{X} \in \mathcal{X}$ for the collection of all node trajectories, i.e. $\mathbf{X} = \{X_1, \ldots, X_N\}$. One typical shape is $\mathbf{X} \in \mathbb{R}^{N\times T\times d}$ if we store them in a 3D tensor. We use a latent directed graph, represented by an adjacency matrix $A \in \{0,1\}^{N\times N}$, where $A_{ij} = 1$ indicates a directed edge from node $i$ to node $j$ (i.e., $i \to j$). We define $E = N^2$ potential edges, enumerated by $e \in \{1,\ldots,E\}$. Moreover, depending on the prediction goals, we separate the time-series $\mathbf{X} = \{\mathbf{x}_1,\ldots,\mathbf{x}_T\}$ into past observations $\mathbf{X}_{\text{past}} = \{\mathbf{x}_1,\ldots,\mathbf{x}_{t_0}\}$, and future targets: $\mathbf{X}_{\text{future}} = \{\mathbf{x}_{t_0+1},\ldots,\mathbf{x}_T\}$ according to time $t_0$.

### 3.2. Structural Inference with VAEs

In this work, we aim to learn a latent graph structure that explains the observed data $\mathbf{X}$ for $N$ interacting nodes. Specifically, we posit a *latent adjacency* $Z$ capturing which nodes influence which others, where $Z$ can be discrete ($\{0,1\}^E$) or continuous ($[0,1]^E$). Each coordinate $Z_e$ denotes the potential presence or weight of edge $e$ among $E$ total edges. We adopt a standard **VAE** approach (Kipf et al., 2018; Alet et al., 2019; Webb et al., 2019; Chen et al., 2021):

We treat $Z$ as a random adjacency in $[0,1]^E$ or $\{0,1\}^E$. A **VAE** comprises an encoder $q_\phi(Z|\mathbf{X})$ and a decoder $p_\theta(\mathbf{X}_{\text{future}} \mid Z, \mathbf{X}_{\text{past}})$, along with a KL term that encourages simpler adjacency. The encoder comes with parameter $\phi$, and it produces parameters $\{\theta_e\}$ for each edge $e$, which can be interpreted as logits in a Bernoulli or Gumbel-softmax distribution: $q_\phi(Z_e = 1|\mathbf{X}) = \sigma(\theta_e)$, or more complex variants. The decoder comes with parameter $\theta$. Given the learned adjacency $Z$ and the past node states $\mathbf{X}_{\text{past}}$, the decoder predicts the future states $\mathbf{X}_{\text{future}}$. We obtain the adjacency from the latent space of the VAE. The VAE objective for structural inference typically includes a part for prediction and KL term:

$$\mathcal{L}_{\text{VAE}}(\phi,\theta) = \underbrace{\mathbb{E}_{q_\phi(Z|\mathbf{X}_{\text{past}})}\big[-\log p_\theta(\mathbf{X}_{\text{future}}|Z,\mathbf{X}_{\text{past}})\big]}_{\text{Prediction Loss}}$$
$$+ \ \beta \ \underbrace{\text{D}_{\text{KL}}\big(q_\phi(Z|\mathbf{X}_{\text{past}}) \,\|\, p(Z)\big),}_{\text{KL Regularizer}} \quad (1)$$

where the prediction loss encourages the learned adjacency to be informative about the next-step states or about reconstructing the entire sequences, and the KL term encourages

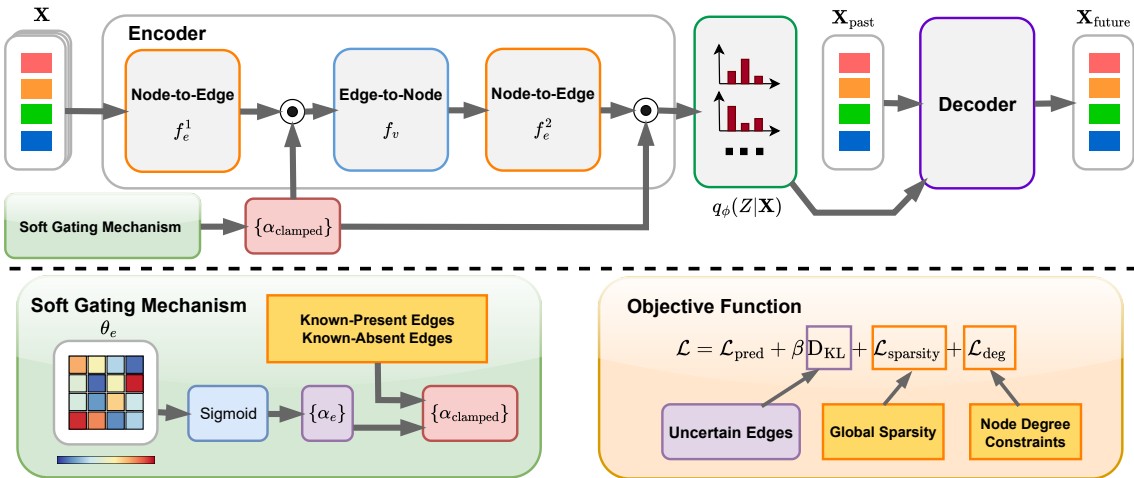

*Figure 1.* **(Above)** The overview of the whole pipeline. **(Bottom)** Details about the soft gating and objective functions in this work.

$q_\phi(Z|\mathbf{X}_{\text{past}})$ to remain close to a prior $p(Z)$, typically pushing for a simpler or more compressed adjacency. The parameter $\beta$ balances how strongly we emphasize compression against prediction, drawing parallels with the information bottleneck principle (Tishby et al., 1999).

While existing methods infer fully unknown adjacency, integrating partial prior knowledge poses challenges: forcibly overwriting edges disrupts gradient flow, and fixing edges to deterministic values contradicts latent uncertainty assumptions, creating conflicting KL signals. SGSI resolves this via soft gating, which seamlessly enforces constraints while preserving gradients and distributional consistency.

### 3.3. Partial Knowledge and Adjacency Constraints

Real-world systems exhibit diverse edges, including known, constrained, and uncertain, yet existing methods treat all edges as uniformly latent, causing conflicts. SGSI integrates partial knowledge (known present/absent edges, sparsity, node degrees) to resolve inconsistencies and enhance inference accuracy. We refer to such knowledge as partial or prior adjacency constraints in the following forms:

**Known-Present Edges:** A subset $\mathcal{E}^+ \subseteq \{(i \to j) \mid 1 \leq i \neq j \leq N\}$ of edges is deemed definitely present. For instance, certain couplings are physically guaranteed to exist (e.g., a gravitational link in a solar system), or certain agent-to-agent channels are known to be always active. We want to ensure any latent adjacency variable $Z_{ij}$ for $(i \to j) \in \mathcal{E}^+$ is effectively "on" (or near 1 in a continuous relaxation).

**Known-Absent Edges:** Another subset $\mathcal{E}^- \subseteq \{(i \to j) \mid 1 \leq i \neq j \leq N\}$ is known to be absent or forbidden. This may arise if domain rules forbid certain interactions (e.g., node $i$ cannot influence node $j$ under the laws of the system), or if data from external knowledge confirms no direct link. Thus, for $(i \to j) \in \mathcal{E}^-$, the adjacency variable $Z_{ij}$ should be near 0 or exactly zero in a discrete adjacency.

**Sparsity or Global Edge Constraints:** In many settings, the number of active edges in a system is expected to be small (e.g., a physical system with local interactions only, or a network known to be mostly disjoint). Alternatively, one might only have an approximate fraction of edges $\rho$ that remain active. This can be enforced via a penalty on the sum of gating values or a more rigid constraint on the total number of edges allowed.

**Node-Degree Constraints:** Certain nodes might be known "hubs", or they might have out-degree (or in-degree) capped by a known property. For example, in a system where each node can only maintain three outgoing connections due to bandwidth limits, or a node might have exactly two neighbors in a ring-like structure. We might penalize or clamp the out-degree $\sum_j Z_{ij}$ to match a target degree $deg_i$.

A naive way to incorporate known edges is to forcibly overwrite the VAE's Sigmoid or Gumbel-softmax outputs for $(i \to j) \in \mathcal{E}^+$ with 1 or for $\mathcal{E}^-$ with 0. However, this often leads to in-place modification conflicts in modern autograd frameworks, as the same tensor is needed for gradient computations. Moreover, if certain edges are forcibly set to 0 or 1 without adjusting the KL term (or removing them from the latent distribution), the prior-latent assumption that all edges are random is violated. The model may receive contradictory signals: on one hand, the KL encourages a distribution on every edge; on the other hand, we are forcibly clamping some edges to fixed values. To address these issues in a gradient-friendly manner, our approach uses soft gating and is discussed in detail in Section 4.

## 4. Structural Inference with Soft Gating

SGSI builds upon the idea that VAE-based structural inference can be interpreted through the lens of the VIB. In this section, we detail how we embed soft gating within a VAE framework, incorporate partial knowledge, and reconcile it

with the IB principle. Fig. 1 illustrates the overall pipeline.

## 4.1. Learned Gating Overview

We start with a VAE-based structural inference setup where the latent variable $Z$ represents the adjacency among $N$ nodes. Instead of treating each edge as strictly binary, we introduce a continuous gating parameter $\theta_e \in \mathbb{R}$ per potential edge $e \in \{1, \ldots, E\}$. Thus we obtain gating values:

$$\alpha_e = \sigma(\theta_e), \qquad (2)$$

where $\sigma(\cdot)$ is the sigmoid function. Each $\alpha_e \in [0, 1]$ acts as a soft mask, suppressing irrelevant edges ($\alpha_e \to 0$) or activating critical ones ($\alpha_e \to 1$). Besides that, the remaining part of the network is setup as the following:

**Encoder** $q_\phi(Z|\mathbf{X}) = \text{softmax}(f_{enc,\phi}(\mathbf{X}))$, where $f_{enc,\phi}$ is implemented by a GNN that acts on the fully conditioned graph. Given node observations $\mathbf{X} = \{X_1, X_2, \ldots, X_j, \ldots, X_N\}$, it outputs logits $\xi$ for each potential edge of the graph.

$$\text{Node Embedding:} \quad \mathbf{h}_j^{(1)} = f_{\text{embed}}^{(1)}(X_j), \qquad (3)$$

$$\text{Node-to-edge:} \quad \mathbf{h}_{ij}^{(1)} = f_e^1([\mathbf{h}_i^{(1)}, \mathbf{h}_j^{(1)}]), \qquad (4)$$

$$\text{Gating 1:} \quad \mathbf{e}_{ij}^{(1)} = \mathbf{h}_{ij}^{(1)} \odot \alpha_e, \qquad (5)$$

$$\text{Edge-to-node:} \quad \mathbf{h}_j^{(2)} = f_v\left(\sum \mathbf{e}_{ij}^{(1)}\right), \qquad (6)$$

$$\text{Node-to-edge:} \quad \mathbf{h}_{ij}^{(2)} = f_e^{(2)}([\mathbf{h}_i^{(2)}, \mathbf{h}_j^{(2)}]), \qquad (7)$$

$$\text{Gating 2:} \quad \mathbf{e}_{ij}^{(2)} = \mathbf{h}_{ij}^{(2)} \odot \alpha_e, \qquad (8)$$

where $[\cdot, \cdot]$ defines concatenation of two tensors. $f_{\text{embed}}^{(1)}$ is the embedding network for input features, $f_e^{(1)}$ and $f_e^{(2)}$ are node-to-edge message-passing networks, and $f_v$ is the edge-to-node operation. Each edge's message is multiplied by $\alpha_e$, thus "turning on" edges close to 1 and suppressing edges near 0. We then model posterior over edges as

$$q_\phi(Z|\mathbf{X}) = \text{softmax}(\mathbf{e}_{ij}^{(2)}), \qquad (9)$$

where $\phi$ summarizes the parameters of the neural networks in Eqn. 3-8. Similar to approaches from (Kipf et al., 2018; Wang & Pang, 2022; Wang et al., 2023), additional randomization such as Gumbel-Softmax is used.

**Decoder** $p_\theta(\mathbf{X}_{\text{future}}|Z, \mathbf{X}_{\text{past}})$ takes in node states and adjacency $\{\mathbf{e}_{ij}^{(2)}\}$ from Eqn. 9 to predict nodes' subsequent states, which are influenced by edges scaled by $\alpha_e$. Following (Kipf et al., 2018; Wang & Pang, 2022; Wang et al., 2023), we use a message-passing framework:

$$\hat{\mathbf{h}}_j(t+1) = x_j(t) + f_{\text{embed}}^{(3)}\left(\sum_{z_{ij} > 0} \mathbf{z}_{ij} f_{\text{embed}}^{(2)}([x_i(t), x_j(t)])\right), \qquad (10)$$

where $f_{\text{embed}}^{(2)}$ and $f_{\text{embed}}^{(3)}$ are multilayer perceptrons (MLPs) to embed the features. In a standard VAE setting, we also

add a KL or information bottleneck penalty on $\xi$, striking a balance between accurate prediction and minimal adjacency. However, the details of this are often shaped by the choice of prior $p(Z)$. In practice, we simply have $\beta$-weighted KL term (Eqn. 1) for the gating distribution.

## 4.2. Incorporating Partial Knowledge

A central contribution of this work is to allow partial domain knowledge about edges, global sparsity or node degrees to be injected into the gating approach without causing in-place modifications or contradictory distributions. SGSI addresses this via gradient-safe clamping and adaptive regularization.

**Known-Present and Known-Absent Edges.** If an edge $e$ is present according to the domain knowledge or prior knowledge, i.e. $e \in \mathcal{E}^+$, we want $\alpha_e \approx 1$. However, directly overwriting the Sigmoid output $\alpha_e$ in-place can break gradient flow. Instead, we use the following steps :

1. Compute $\alpha_e = \sigma(\theta_e)$ for all edges.
2. Clone $\alpha_e$ into $\alpha_{\text{clamped}}$.
3. Clamp in $\alpha_{\text{clamped}}$: for $e \in \mathcal{E}^+$, set $\alpha_{\text{clamped}}[e] = 1$.
4. Freeze $\xi_e$ for $e \in \mathcal{E}^- \cup \mathcal{E}^+$ (e.g., $\xi_e \to +\infty$ for $\mathcal{E}^+$).
5. Exclude clamped edges from the KL divergence penalty.

Cloning $\alpha_e$ ensures gradient computations use the original $\alpha_e$, while $\alpha_{\text{clamped}}$ enforces domain constraints during forward passes. Skipping KL terms for known edges avoids conflicting signals between the prior and clamped values.

**Global Sparsity.** Beyond edge-by-edge knowledge, we allow global constraints to be inserted via adaptive penalties. Let $\alpha_e$ or the $\alpha_{\text{clamped}}$ be the final gating values for each edge, depending on whether the edges appear in the prior set or not. We define the penalty for sparsity as:

$$\mathcal{L}_{\text{sparsity}} = \lambda_{\text{sparsity}} \cdot \sum_{e=1}^{E} \alpha_e, \qquad (11)$$

encouraging the sum of all gating values to be small. $\lambda_{\text{sparsity}}$ is the weight of the sparsity penalty. Sometimes we want exactly (or approximately) $\rho E$ edges active, where $E$ is the total number of potential edges. For example, $\rho = 0.1$ means 10% edges. We achieve this by comparing the sum of gating $\sum_e \alpha_e$ to $\rho E$:

$$\mathcal{L}_{\text{sparsity}} = \lambda_{\text{sparsity}} \cdot \left|\sum_{e=1}^{E} \alpha_e - \rho E\right|. \qquad (12)$$

This is a soft approach as the model is free to deviate if the data demands a bit more or fewer edges, but it pays a penalty in proportion to how far it strays from $\rho E$.

**Node Degree Constraints.** In many real-world settings, we may know each node $i$ has an exact out-degree $k_i$, or similarly, an in-degree constraint. For example, a node might precisely need 2 out-connections based on domain rules. We incorporate these constraints in a soft manner rather

than forcing exact rewrites, which could lead to in-place modifications and contradictory distributions. Following the notations in Section 3.1, we interpret $\alpha_e$ as how "active" edge $e$ is. If edge $e$ corresponds to $(i \to j)$, then node $i$ is the sender. We have the out-degree and in-degree of $i$:

$$\text{outdeg}(i) = \sum_{j=1}^{N} \alpha_{(i \to j)}. \tag{13}$$

$$\text{indeg}(i) = \sum_{j=1}^{N} \alpha_{(j \to i)}. \tag{14}$$

Depending on whether we have an out-degree or in-degree constraint, we focus on one of these sums or both. Suppose we want node $i$ to have out-degree exactly $k_i$. We can define:

$$\mathcal{L}_{\text{deg}} = \lambda_{\text{deg}} \cdot \sum_{i=1}^{M} \left| \text{outdeg}(i) - k_i \right|, \tag{15}$$

where $M$ represents the set of nodes having out-degree constraints and is a subset of all $N$ nodes. Let $M \subseteq \{1, \dots, N\}$ be the set of nodes with out-degree constraints, and let $\lambda_{\text{deg}}$ be the penalty weight. This term softly enforces $\sum_j \alpha_{i \to j} \approx k_i$, but allows deviations if supported by data (at a penalty cost). For in-degree constraints, replace $\text{outdeg}(i)$ with $\text{indeg}(i)$. If some nodes have out-degree constraints and others have in-degree constraints, handle each node independently and sum the corresponding terms.

## 4.3. Objective Function

The training objective balances prediction accuracy, model complexity, and adherence to prior knowledge. The **prediction loss** is the same as that in Eqn. 1:

$$\mathcal{L}_{\text{pred}} = -\log p_\theta(\mathbf{X}_{\text{future}} | \alpha_{\text{clamped}}, \mathbf{X}_{\text{past}}). \tag{16}$$

However, the **KL term** may vary depending on the prior knowledge. If certain edges are truly fixed, we remove them from the KL computation, avoiding contradictory signals. This is straightforward if we treat those edges as having zero entropy (delta function) in the distribution, or if we skip them entirely from the random portion. For partially known constraints (like node degrees), we keep the normal KL but apply the external penalty. This synergy ensures a stable co-existence of variational inference and domain constraints. Therefore, based on the prior knowledge we have, we update the $\text{D}_{\text{KL}}$ term in Eqn. 1 accordingly:

$$\text{D}_{\text{KL}} = \sum_{e \in \mathcal{U}} \text{KL}\left( q_\phi\left(\xi_e \mid \mathbf{X}\right) \, \big\| \, p(\xi_e) \right), \tag{17}$$

where $\mathcal{U} = \{1, \dots, E\} \setminus \left( \mathcal{E}^+ \cup \mathcal{E}^- \right)$ represents the set of uncertain edges, $\xi$ denotes the variables in latent space. We then combine the penalties for known degrees or global sparsity with the prediction loss and $\beta$-weighted KL:

$$\mathcal{L} = \mathcal{L}_{\text{pred}} + \beta \, \text{D}_{\text{KL}} + \mathcal{L}_{\text{sparsity}} + \mathcal{L}_{\text{deg}}. \tag{18}$$

## Choosing Hyperparameters.

- **KL Weight** $\beta$: Balances prediction accuracy against model complexity. We typically begin with $\beta = 1.0$ (a balanced trade-off). For very sparse or low-data scenarios, we may reduce $\beta$ (e.g., 0.1) to prioritize prediction. Conversely, in settings demanding strong regularization or interpretability, $\beta$ can be increased (e.g., 2.0 or 5.0).

- **Sparsity Weight** $\lambda_{\text{sparsity}}$: Controls adherence to global sparsity. For exact sparsity ($\sum \alpha_e \approx \rho E$), set $\lambda_{\text{sparsity}} \in [0.1, 1.0]$. For soft encouragement, use smaller values ($10^{-3}$ to $10^{-2}$).

- **Degree Penalty** $\lambda_{\text{deg}}$: Similarly tested in $[10^{-4}, 10^{-2}]$. Larger $\lambda_{\text{deg}}$ enforces node-degree constraints more strictly, while smaller values let data cues override the prior knowledge. We pick a value based on a small validation set or best AUROC for edge recovery.

In practice, we find a brief grid search or Bayesian optimization on $(\beta, \lambda_{\text{sparsity}}, \lambda_{\text{deg}})$ is sufficient to find hyperparameters that yield both high-fidelity adjacency and stable training for the experiments in this work.

## 4.4. Interpretation via VIB

SGSI formalizes the IB principle by explicitly separating known and uncertain edges in the latent adjacency $Z$. Let $\mathbf{X}$ denote observed node trajectories, and $Z$ represent the latent graph with three disjoint subsets:

- $\mathcal{E}^+$: Known-present edges (fixed to 1).
- $\mathcal{E}^-$: Known-absent edges (fixed to 0).
- $\mathcal{U}$: Uncertain edges (learned from data).

**Posterior and Prior Factorization**. The encoder $q_\phi(Z|\mathbf{X})$ and prior $p(Z)$ factorize as:

$$q_\phi(Z|\mathbf{X}) =$$
$$\prod_{e \in \mathcal{U}} q_\phi(Z_e|\mathbf{X}) \cdot \prod_{e \in \mathcal{E}^+} \delta(Z_e = 1) \cdot \prod_{e \in \mathcal{E}^-} \delta(Z_e = 0), \tag{19}$$
$$p(Z) = \prod_{e \in \mathcal{U}} p(Z_e) \cdot \prod_{e \in \mathcal{E}^+} \delta(Z_e = 1) \cdot \prod_{e \in \mathcal{E}^-} \delta(Z_e = 0), \tag{20}$$

where $\delta(\cdot)$ enforces deterministic values for known edges. This factorization ensures:

- Known edges ($\mathcal{E}^+, \mathcal{E}^-$) are excluded from the stochastic portion of $Z$, avoiding contradictory signals between data and prior knowledge.
- Uncertain edges ($\mathcal{U}$) remain random, allowing the model to learn their distributions.

**KL Divergence Simplification**. The KL divergence be-

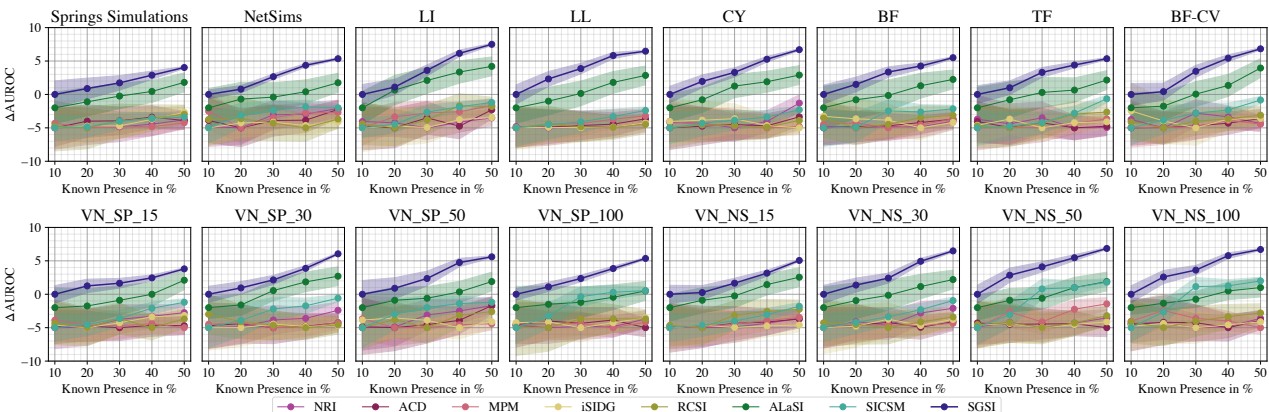

*Figure 2.* The average $\Delta$ AUROC values of ten runs of baselines and SGSI on the datasets with different percentages of known present edges. The shadings represent the standard deviations.

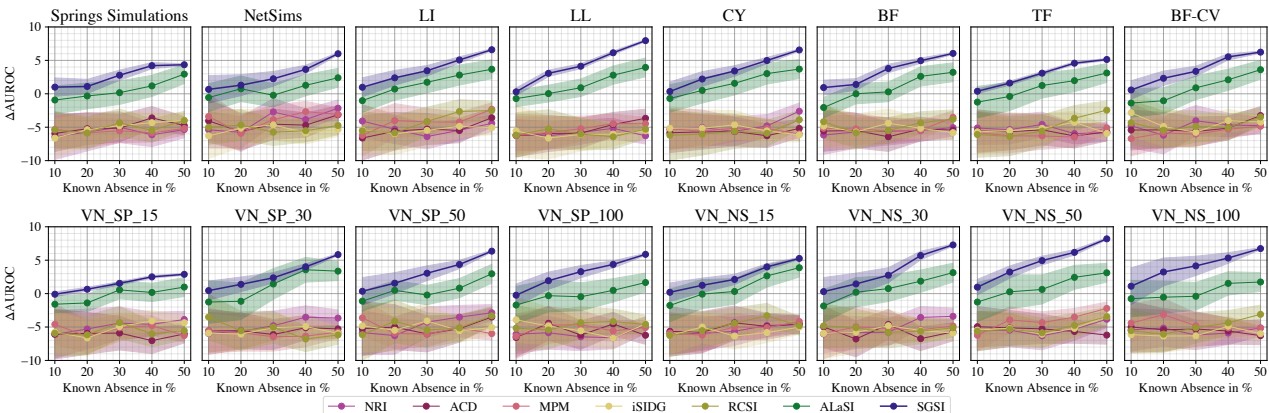

*Figure 3.* The average $\Delta$AUROC values of ten runs of baselines and SGSI on the datasets with different percentages of known absent edges. The shadings represent the standard deviations.

tween posterior and prior reduces to:

$$D_{\mathrm{KL}}\big(q_\phi(Z|\mathbf{X})\|p(Z)\big) = \sum_{e \in \mathcal{U}} D_{\mathrm{KL}}\big(q_\phi(Z_e|\mathbf{X})\|p(Z_e)\big),$$

(21)

since $D_{\mathrm{KL}}(\delta\|\delta) = 0$ for known edges, aligning with IB:

- **Compression** ($I(X;Z)$): The KL term penalizes extraneous information stored in uncertain edges. Known edges consume no "bits" due to their deterministic nature.
- **Prediction** ($I(Z;Y)$): The prediction loss ensures $Z$ retains edges critical for predicting $\mathbf{X}_{\mathrm{future}}$.

**Theoretical Implications**. By delta-encoding known edges, SGSI achieves two key properties:

1. **No Contradictory Signals**: Known edges are excluded from the random latent space, preventing conflicts between prior assumptions (e.g., a uniform prior) and domain knowledge.
2. **Freed Capacity**: The model allocates information-theoretic "bits" only to uncertain edges, optimizing the

IB trade-off:

$$\min_{Z}\Big[\ \underbrace{I(X;Z)}_{\text{Uncertain edges}}\ -\beta\underbrace{I(Z;Y)}_{\text{Prediction}}\Big].$$

The soft gating $\alpha_e = \sigma(\theta_e)$ directly implements this IB-driven compression as the known edges ($\alpha_e = 1$ or $0$) are deterministic, contributing nothing to $I(X;Z)$. Besides, uncertain edges ($\alpha_e \in (0,1)$) are regularized by the KL term and sparsity penalties, limiting their information content.

## 5. Experimental Results

In this section, we evaluate the proposed soft gating approach on various datasets. We aim to investigate accuracy, graph quality, robustness, flexibility and the information bottleneck perspective of SGSI.

### 5.1. General Settings

**Datasets.** Our study first evaluates the SGSI model on two established structural inference datasets: the Spring

Simulations dataset (Kipf et al., 2018), which simulates dynamic interactions of balls connected by springs within a symmetric setting, and the NetSim dataset (Smith et al., 2011b), which consists of simulated blood-oxygen-level-dependent imaging data from various brain regions in an asymmetric network. Both datasets include 10 nodes, with Spring Simulations offering four-dimensional features and NetSim one-dimensional features at each timestep, initially sampled at 49 regular intervals.

Furthermore, we examined six directed synthetic biological networks (Linear, Linear Long, Cycle, Bifurcating, Trifurcating, and Bifurcating Converging) as outlined in (Pratapa et al., 2020), with abbreviations LI, LL, CY, BF, TF and BF-CV, respectively. These networks simulate developmental trajectories in differentiating cells using BoolODE (Pratapa et al., 2020), capturing one-dimensional mRNA expression levels over 49 timesteps.

We also incorporated data from the StructInfer Benchmark (Wang et al., 2024), focusing on 'Vascular Networks' (VN) with node counts ranging from 15 to 100. These datasets, named under the categories Springs (SP) and Net-Sims (NS), were selected for their complex and varying underlying graph structures, providing a robust platform to validate the efficacy of the SGSI model.

**Baselines.** To evaluate the performance of SGSI, we compared it against a suite of state-of-the-art models:

- NRI (Kipf et al., 2018): a VAE-based model for unsupervised relational inference.
- MPM (Chen et al., 2021): employs a VAE framework with a relational interaction mechanism and spatio-temporal message passing.
- ACD (Löwe et al., 2022): utilizes shared dynamics to infer causal relations within datasets.
- iSIDG (Wang & Pang, 2022): iteratively refines adjacency matrices to enhance directional inference.
- RCSI (Wang et al., 2023): integrates reservoir computing for efficient structural inference.
- ALaSI (Wang & Pang, 2023): a structural inference method based on deep active learning.
- SICSM (Wang & Pang, 2024a): a conjoined state space model for complex input data modeling.

Since NRI, MPM, ACD, iSIDG and RCSI do not provide an official way of integrating prior knowledge, we just rewrite the adjacency matrix in their latent spaces to integrate known present/absent edges. ALaSI can have known present/absent edges integrated by default, and SICSM can only have known present edges integrated. Yet we did not find a possible way of having global sparsity and node degree constraints to be integrated into any of these baselines.

**Metrics.** Similar to prior work, the comparative effectiveness of these methods was quantitatively assessed using the area under the receiver operating characteristic (AUROC) curve, focusing on the accuracy of the inferred adjacency matrix. All experiments were conducted on a single NVIDIA Ampere 40GB HBM graphics card, paired with 2 AMD Rome CPUs (32 cores@2.35 GHz). For more details about the experiments, please refer to Appendix E.

## 5.2. Varying Amount of Known Edges

To assess how partial knowledge impacts structural inference, we evaluate each method under varying percentages of known-absent (10%–50%) and known-present (10%–50%) edges, and demonstrate the results in Fig. 2 and Fig. 3. We measure improvements in structural inference quality via $\Delta$AUROC, representing the change in AUROC relative to the same method without any prior knowledge. Higher $\Delta$AUROC indicates a stronger benefit from partial adjacency constraints. Across all baselines that rely on naive overwrites, improvements in $\Delta$AUROC typically are negative. The forcibly pinned edges begin to contradict the latent distributions expected by these frameworks. Methods that forcibly zero out edges often find it easier to eliminate false positives, leading to negative $\Delta$AUROC gains. For known-present edges, certain baselines, especially ones that strongly rely on a random adjacency distribution such as NRI, ACD, iSIDG and RCSI, struggle to fully accommodate forced 1s if that set grows large.

While we leave global sparsity or node-degree constraints for the next section, it is already clear that naive in-place overwrites for known edges do not scale gracefully. In contrast, SGSI with the soft gating approach, which skips or reduces KL for clamped edges, consistently reassigns capacity to the uncertain edges without harming gradient flow. This synergy yields higher $\Delta$AUROC at higher knowledge fractions, which is a point we underscore once additional constraints come into play. These results confirm that most baselines (NRI, MPM, ACD, iSIDG, RCSI) can not handle any partial adjacency knowledge via overwrites and exhibit remarkable negative gains at higher known-edge fractions. ALaSI and SICSM integrate partial knowledge more directly, yet remain bound by their inability to handle large sets of forced edges or forcibly absent edges (in the case of SICSM). Overall, this points to the need for a more systematic, gradient-friendly method that properly excludes known edges from the random portion of the adjacency, which is a motivation leading directly to our soft gating solution.

## 5.3. Global Sparsity and Node Degree Constraints

In Table 1, we compare the baselines, which lack any official support for partial knowledge, to SGSI under four variants: (1) SGSI w. Sparsity: A mild penalty on the total adjacency sum against the known sparsity. (2) SGSI w. In-deg.: Each node's in-degree is softly constrained by domain knowledge.

*Table 1.* Baseline AUROC results without prior knowledge compared to SGSI variants: SGSI w. Sparsity (global sparsity), SGSI w. In-deg. (known in-degrees), SGSI w. Out-deg. (known out-degrees), and SGSI w. Both-deg. (both in- and out-degrees).

| Method | Dataset | | | | | | | |
|---|---|---|---|---|---|---|---|---|
| | VN_SP_15 | VN_SP_30 | VN_SP_50 | VN_SP_100 | VN_NS_15 | VN_NS_30 | VN_NS_50 | VN_NS_100 |
| NRI | $94.58_{\pm 0.01}$ | $95.12_{\pm 0.01}$ | $94.65_{\pm 0.02}$ | $89.17_{\pm 0.02}$ | $90.31_{\pm 0.01}$ | $74.64_{\pm 0.04}$ | $69.78_{\pm 0.03}$ | $68.80_{\pm 0.02}$ |
| MPM | $96.56_{\pm 0.01}$ | $89.71_{\pm 0.04}$ | $85.07_{\pm 0.02}$ | $84.56_{\pm 0.03}$ | $91.18_{\pm 0.01}$ | $83.37_{\pm 0.03}$ | $72.66_{\pm 0.04}$ | $70.34_{\pm 0.03}$ |
| ACD | $94.34_{\pm 0.01}$ | $93.73_{\pm 0.01}$ | $87.54_{\pm 0.03}$ | $90.49_{\pm 0.03}$ | $80.32_{\pm 0.02}$ | $65.36_{\pm 0.06}$ | $69.01_{\pm 0.03}$ | $68.72_{\pm 0.03}$ |
| iSIDG | $96.59_{\pm 0.02}$ | $95.66_{\pm 0.01}$ | $95.72_{\pm 0.02}$ | $85.07_{\pm 0.02}$ | $91.20_{\pm 0.02}$ | $78.08_{\pm 0.06}$ | $73.68_{\pm 0.02}$ | $68.81_{\pm 0.02}$ |
| RCSI | $97.03_{\pm 0.01}$ | $95.31_{\pm 0.01}$ | $94.48_{\pm 0.02}$ | $90.72_{\pm 0.02}$ | $91.53_{\pm 0.02}$ | $82.27_{\pm 0.04}$ | $74.08_{\pm 0.02}$ | $70.29_{\pm 0.03}$ |
| ALaSI | $96.27_{\pm 0.02}$ | $95.30_{\pm 0.01}$ | $96.43_{\pm 0.02}$ | $91.05_{\pm 0.03}$ | $89.03_{\pm 0.04}$ | $80.35_{\pm 0.06}$ | $75.31_{\pm 0.04}$ | $73.54_{\pm 0.04}$ |
| SICSM | $97.70_{\pm 0.03}$ | $96.88_{\pm 0.02}$ | $90.24_{\pm 0.03}$ | $94.78_{\pm 0.04}$ | $95.38_{\pm 0.03}$ | $94.21_{\pm 0.05}$ | $79.24_{\pm 0.05}$ | $80.21_{\pm 0.07}$ |
| SGSI | $95.29_{\pm 0.02}$ | $92.76_{\pm 0.01}$ | $90.92_{\pm 0.03}$ | $88.17_{\pm 0.04}$ | $93.25_{\pm 0.04}$ | $88.06_{\pm 0.03}$ | $79.05_{\pm 0.03}$ | $77.90_{\pm 0.02}$ |
| SGSI w.Sparsity. | $97.80_{\pm 0.02}$ | $96.90_{\pm 0.01}$ | $96.43_{\pm 0.03}$ | $93.91_{\pm 0.04}$ | $96.26_{\pm 0.03}$ | $94.01_{\pm 0.02}$ | $82.92_{\pm 0.02}$ | $81.54_{\pm 0.02}$ |
| SGSI w.In-deg. | $97.42_{\pm 0.02}$ | $96.82_{\pm 0.02}$ | $95.90_{\pm 0.03}$ | $93.05_{\pm 0.04}$ | $95.83_{\pm 0.02}$ | $94.76_{\pm 0.02}$ | $80.80_{\pm 0.03}$ | $80.34_{\pm 0.02}$ |
| SGSI w.Out-deg. | $97.43_{\pm 0.02}$ | $96.80_{\pm 0.02}$ | $95.87_{\pm 0.03}$ | $93.01_{\pm 0.04}$ | $95.62_{\pm 0.02}$ | $94.87_{\pm 0.02}$ | $81.05_{\pm 0.03}$ | $80.55_{\pm 0.02}$ |
| SGSI w.Both-deg. | $\mathbf{97.90}_{\pm 0.02}$ | $\mathbf{96.95}_{\pm 0.03}$ | $\mathbf{96.46}_{\pm 0.04}$ | $\mathbf{94.81}_{\pm 0.03}$ | $\mathbf{96.32}_{\pm 0.03}$ | $\mathbf{95.00}_{\pm 0.02}$ | $\mathbf{83.56}_{\pm 0.03}$ | $\mathbf{82.08}_{\pm 0.02}$ |

(3) SGSI w. Out-deg.: Similarly, each node's out-degree is guided by known limits or exact values. (4) SGSI w. Both-deg.: Combining in-degree and out-degree constraints for all nodes. The table shows average AUROC results of ten runs each on multiple datasets.

As we can see from the table, having *sparsity* typically raises final AUROC by 2–6% (e.g., from 92.76% to 96.90% on VN_SP_30), reflecting how reducing unnecessary edges lets the model focus on essential interactions. Meanwhile, *in-/out-degree* constraints further boost accuracy wherever domain rules about node connectivity align with the true graph. For instance, specifying in-degrees on VN_SP_50 increases AUROC from 90.92% to 95.90%. Enforcing both in- and out-degrees usually yields the top or near-top scores; on VN_NS_50, performance jumps from 79.05% (unconstrained) to 83.56%.

Overall, these results confirm that SGSI not only integrates partial adjacency knowledge on edge presence/absence but also extends smoothly to global sparsity and node-degree constraints. The method can gracefully deviate if data demands it, leading to *state-of-the-art* performance across diverse domains and underscoring the flexibility of SGSI.

### 5.4. Ablation Studies

In Fig. 4, we show the average loss curves of SGSI and its two variants: without KL regularization of uncertain edges and without cloning of the gating variables, on VN_SP_50 dataset. As shown, Default SGSI converges to the lowest final loss with minimal oscillation, whereas failing to skip known edges in the KL or omitting the cloning step each lead to higher loss and greater instability. Moreover, both of the two variants yield lower or "matching" adjacency accuracy relative to each other and well below default SGSI's performance. From an IB perspective, not skipping known

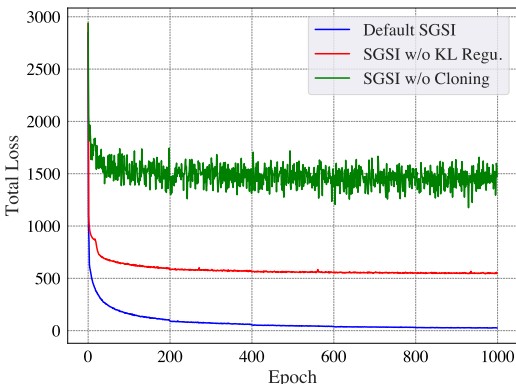

*Figure 4.* Average loss curves for three variants of SGSI: Default SGSI (blue), SGSI without KL regularization of uncertain edges (SGSI w/o KL Reg., red), and SGSI without cloning of the gating variables (SGSI w/o Cloning, green).

edges in the KL term (the red curve) forces the encoder to "spend bits" on edges that are forcibly pinned, while not cloning gating (the green curve) disrupts stable gradient updates altogether. This aligns with the broader rationale behind VAE-based structural inference (e.g., ACD (Löwe et al., 2022), iSIDG (Wang & Pang, 2022), RCSI (Wang et al., 2023)), in which the Variational Information Bottleneck concept helps explain why prioritizing uncertain edges yields more effective adjacency discovery. Hence, these results strongly validate the soft-gating pipeline, in which both skipping known edges' KL costs and cloning before clamping are vital for stable optimization and maximal benefit from partial knowledge.

### 5.5. Robustness to Inaccurate Prior Knowledge

SGSI leverages soft penalties and cloned gating to integrate partial knowledge while preserving flexibility. Because these constraints are applied softly, via mild penalties and by clamping only a cloned copy of the gating vector,

*Table 2.* Performance under Various Perturbations on VN_SP_100 and VN_NS_100. "Raw" denotes the unperturbed baseline. "F. 20% known-present" and "F. 50% known-present" indicate that 20% and 50% of the known-present edges (within a 30% prior knowledge set) are randomly flipped, respectively; similarly for "F. 20% known-absent" and "F. 50% known-absent." "10%/20% E. Sparsity" represent 10%/20% errors on the global sparsity, while "10%/20% E. In-Deg." and "10%/20% E. Out-Deg." denote errors on in-degree and out-degree estimates, respectively.

| | Raw | F. 20% KP | F. 50% KP | F. 20% KA | F. 50% KA | 10% E. Sparsity | 20% E. Sparsity | 10% E. In-Deg. | 20% E. In-Deg. | 10% E. Out-Deg. | 20% E. Out-Deg. |
|---|---|---|---|---|---|---|---|---|---|---|---|
| VN_SP_100 | 88.17 | 90.03 | 89.70 | 91.45 | 90.16 | 92.61 | 91.08 | 92.70 | 90.94 | 92.66 | 90.82 |
| VN_NS_100 | 77.90 | 80.18 | 78.76 | 81.06 | 79.68 | 80.09 | 79.35 | 79.36 | 78.18 | 79.80 | 78.23 |

the model can deviate from incorrect priors when the data strongly contradicts them. In SGSI, the soft gating mask is applied after the Node-to-Edge operation (but not after Edge-to-Node), which helps preserve residual connectivity and prevents over-reliance on potentially flawed edges.

To further validate this, we conducted "noisy prior" experiments on VN_SP_100 and VN_NS_100, where we randomly flipped 20% or 50% of the known-present/absent edges (with the overall prior knowledge set to 30%) and introduced 10% or 20% errors in global sparsity and degree constraints. Table 2 summarizes our preliminary results.

These results indicate that even with moderate noise, SGSI remains significantly more accurate than a no-knowledge baseline with at least 1 2% margin. Although performance gains naturally diminish as noise increases, the model robustly leverages available prior knowledge without catastrophic failure. For more experimental results, including on real-world data, downstream tasks, and time complexity, please refer to Appendix E.

## 6. Conclusion

This paper introduces SGSI, a soft-gating approach for VAE-based structural inference that enables partial adjacency knowledge (known-present/absent edges, global sparsity and node degree constraints) to be integrated smoothly and without in-place gradient conflicts. Empirically, SGSI surpasses existing baselines in structural fidelity, particularly at higher fractions of known edges. By skipping KL on forcibly clamped edges and cloning before clamping, SGSI preserves stable gradient flow and effectively "frees capacity" for uncertain edges, whose insights align well with an information bottleneck perspective.

Beyond relational discovery, these ideas of selectively excluding known factors from a latent distribution can generalize to broader contexts. Potential future directions include dynamic graph inference, reinforcement learning where partial environment transitions are known, or even multimodal data fusion tasks such as integrating known sensor links with learned ones. In each domain, soft gating and careful KL management may unlock synergy between domain constraints and data-driven latent representations.

## Impact Statement

This work's soft-gating approach to knowledge-aware structural inference enables more trustworthy and transparent graph discovery by seamlessly integrating partial domain expertise. Beyond scientific modeling, it may influence fields like social network analysis or medical diagnostics, where partial relationships are known a prior. By clarifying latent structures in these sensitive domains, the technique has potential ethical benefits—reducing spurious inferences and fostering safer, more explainable models. However, it also carries the societal consequence that misuse or over-reliance on domain knowledge could amplify biases in the data, underscoring the need for careful, context-informed deployment.

## Acknowledgements

The experiments presented in this paper were carried out using the HPC facilities of the University of Luxembourg (Varrette et al., 2022) (see `hpc.uni.lu`). Besides that, author Jun Pang acknowledges financial support of the Institute for Advanced Studies of the University of Luxembourg through an Audacity Grant (AUDACITY-2021).

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

# Appendix of Guided Structural Inference: Leveraging Priors with Soft Gating Mechanisms

## A. More Details about SGSI

### A.1. Sparsity Penalty

**Alternative.**  Besides the form Eqn. 11-12, we could also do an L2 penalty:

$$\mathcal{L}_{\text{sparsity}} \Big(\sum_e \alpha_e \ - \ \rho E\Big)^2, \tag{22}$$

which more strongly penalizes large deviations. Either L1 or L2 works in practice, though L1 typically yields sparser solutions.

**Combining with Known Edges.**  If a subset $\mathcal{E}^+$ of edges is forcibly clamped to $\alpha_e = 1$, these edges are effectively non-negotiable. Since we want the final fraction $\rho E$ to be in addition to these forced edges, we do:

$$\sum_{\text{unknown}} \alpha_e \ = \ \sum_{e \notin \mathcal{E}^+} \alpha_e, \quad \mathcal{L}_{\text{sparsity}} \ = \ \lambda_{\text{sparsity}} \Big| \sum_{\text{unknown}} \alpha_e - \big(\rho E - |\mathcal{E}^+|\big) \Big|. \tag{23}$$

That ensures the known edges are not "counted" against the target fraction.

**Practical Considerations.**  Besides the functioning and technical parts mentioned above, we provide practical instructions for the implementation of sparsity penalty:

**A. Initialization:**  If gating parameters $\theta_e$ are initialized around $-2.0$ or $-4.0$, the initial $\sigma(\theta_e)$ is near 0. This already starts the network off with mostly "off" edges, which can help if we strongly suspect sparsity.

**B. Tuning:** The hyperparameter $\lambda_{\text{sparsity}}$ is crucial. If we see the final adjacency ignoring the fraction $\rho$, increase $\lambda_{\text{sparsity}}$. If it prunes edges too aggressively, decrease $\lambda_{\text{sparsity}}$.

**C. Influence on KL:**  The KL or IB penalty also encourages smaller gating on uncertain edges. Combining the KL with an explicit global-sparsity penalty can lead to very pruned solutions, so please watch out for over-pruning if both $\beta$ (KL weight) and $\lambda_{\text{sparsity}}$ are large.

### A.2. Node Degree Penalty

**Maximum Degree Penalty.**  As we only mentioned the exact degree penalty in Section 4.2, it is feasible of setting up penalty for maximum node degrees. Suppose we want node $i$ to have out-degree at most $k_i$. We can define:

$$\mathcal{L}_{\text{deg}} = \lambda_{\text{deg}} \sum_{i=1}^{N} \Big[ \max\big(0, \ \text{outdeg}(i) - k_i\big) \Big]. \tag{24}$$

This means as soon as $\text{outdeg}(i)$ is above $k_i$, we pay a cost proportional to the surplus. If $\text{outdeg}(i) \leq k_i$, penalty is zero. Similarly, for in-degree, we replace $\text{outdeg}(i)$ with $\text{indeg}(i)$.

**Combining with Known Edges.**  If node $i$ has some edges forcibly set to 1 (known present) or 0 (known absent), they naturally contribute to or omit from the sum in $\text{outdeg}(i)$. That means:

- Known Present edges from $i \to j$ each add 1 to $i$'s out-degree. If the domain says "these edges must exist," we might decide not to penalize them if they push out-degree above $k_i$. Alternatively, we still keep them in the sum. If the node has to not exceed $k_i$ total, known edges reduce the "budget" for the uncertain edges.

- Known Absent edges contribute 0. No penalty conflict arises there, since they do not increase out-degree.

**Practical Considerations.** Besides the functioning and technical parts mentioned above, we provide practical instructions for the implementation of node degree penalty:

**Tuning the Degree Penalty.** Hyperparameter $\lambda_{\text{deg}}$ controls how strongly the network tries to meet node-degree constraints:

- If $\lambda_{\text{deg}}$ is too low, the model may ignore the desired degree in favor of better reconstruction.

- If $\lambda_{\text{deg}}$ is too high, we risk overshadowing the data-driven cues, forcing all or most nodes to match $k_i$ even if that mismatches the observation signals.

We start with a small value (like $10^{-3}$ or $10^{-2}$) and see how the final out-degree distribution looks, adjusting from there.

**Interplay with KL/IB:** The IB or KL cost also encourages compression, possibly making edges smaller or fewer. Combining it with node-degree constraints might reinforce each other or conflict if the data strongly wants more edges for certain nodes.

**Interplay with Global Sparsity:** We can have both a global L1 penalty (pushing overall fewer edges) and a node-degree penalty. Typically, each node's out-degree constraint is more specific, while the global penalty is broad.

If the node-degree sum across all nodes is contradictory to the desired global fraction $\rho E$, the model will struggle. For instance, if each node is forced to have out-degree 2 but $\rho$ implies only $\approx 10\%$ edges overall, yet $2N$ might exceed that fraction. Ensuring consistency in the constraints is crucial.

## A.3. Implementation of SGSI

The overall pipeline of SGSI can be described in Algorithm 1. The implementation pseudocode of encoder can be found in Algorithm 2. The decoder of SGSI is same as the `MLPDecoder` of NRI (Kipf et al., 2018).

---

**Algorithm 3** Two-Layer MLP with ELU and Batch Normalization

**Require:** Input dim $n_{\text{in}}$, hidden dim $n_{\text{hid}}$, output dim $n_{\text{out}}$, dropout rate $d$
**Ensure:** Output tensor $\mathbf{y}$ with shape $[\text{batch}, \text{seq}, n_{\text{out}}]$
 1: **Initialize parameters:**
 2: $W_1 \leftarrow \text{XavierNormal}(n_{\text{in}}, n_{\text{hid}})$           ▷ Layer 1 weights
 3: $b_1 \leftarrow 0.1$           ▷ Layer 1 bias
 4: $W_2 \leftarrow \text{XavierNormal}(n_{\text{hid}}, n_{\text{out}})$           ▷ Layer 2 weights
 5: $b_2 \leftarrow 0.1$           ▷ Layer 2 bias
 6: $\gamma \leftarrow 1, \ \beta \leftarrow 0$           ▷ Batch norm params
 7: **Forward x**           ▷ Input shape: $[\text{batch}, \text{seq}, n_{\text{in}}]$
 8: $\mathbf{h} \leftarrow \text{ELU}\big(\mathbf{x}W_1^\top + b_1\big)$
 9: $\mathbf{h} \leftarrow \text{Dropout}(\mathbf{h}, d)$           ▷ Active only in training
10: $\mathbf{h} \leftarrow \text{ELU}\big(\mathbf{h}W_2^\top + b_2\big)$
11: $\mathbf{h} \leftarrow \text{BatchNorm}\big(\mathbf{h}, \gamma, \beta\big)$           ▷ Sequence-wise
12: **Return: h**

---

**Algorithm 4** Node-to-Edge Mapping with Gating

**Require:** Node embeddings $\mathbf{V} \in \mathbb{R}^{b \times N \times d}$, relation matrices $\mathbf{R}_{\text{rec}}, \mathbf{R}_{\text{send}} \in \mathbb{R}^{E \times N}$, gating vector $\mathbf{g} \in [0,1]^E$
**Ensure:** Gated edge features $\mathbf{E}_{\text{masked}} \in \mathbb{R}^{b \times E \times 2d}$
 1: Transpose $\mathbf{V}$ to $\mathbf{V}^\top \in \mathbb{R}^{b \times d \times N}$
 2: Expand $\mathbf{R}_{\text{send}}^\top, \mathbf{R}_{\text{rec}}^\top$ to batch size $b$
 3: Compute $\mathbf{S} \leftarrow \mathbf{V}^\top \times \mathbf{R}_{\text{send}}^\top$
 4: Compute $\mathbf{R} \leftarrow \mathbf{V}^\top \times \mathbf{R}_{\text{rec}}^\top$
 5: Concatenate $\mathbf{E} \leftarrow \text{Concat}(\mathbf{S}, \mathbf{R})$ along feature dim
 6: Apply gating: $\mathbf{E}_{\text{masked}} \leftarrow \mathbf{E} \odot (\mathbf{g} \otimes \mathbf{1}_{2d})$
 7: **Return: $\mathbf{E}_{\text{masked}}$**

---

---

**Algorithm 1** Soft-Gated Structural Inference (SGSI) Training

---

**Require:** Dataset $\{\mathbf{X}\}$, known edges $\mathcal{E}^+$ (present), $\mathcal{E}^-$ (absent), node-degree/sparsity constraints (optional), hyperparams $(\beta, \lambda_{\mathrm{deg}}, \lambda_{\mathrm{sparsity}}, \dots)$

**Ensure:** Learned parameters $(\phi, \theta)$ and final gating $\{\alpha_e\}$

1: **Initialize** encoder parameters $\phi$, decoder parameters $\theta$, gating logits $\{\theta_e\}_{e=1}^E$
2: **for** each training iteration **do**
3:     Sample mini-batch $\mathbf{X}_{\mathrm{batch}} = \{X_j\}_{j=1}^N$ from the dataset
4:     *(A) Encoder Forward*
5:         Compute $\theta_e = f_{\mathrm{enc}}(\mathbf{X}_{\mathrm{batch}})$ (e.g., via MLP or GNN)
6:     **Obtain soft adjacency gating (size $E$):** $\alpha \leftarrow \sigma(\theta)$
7:     **Cloning/Clamping**:
8:         $\alpha_{\mathrm{clamped}} \leftarrow \alpha.\mathrm{clone}()$
9:         **for** $e \in \mathcal{E}^+$: $\alpha_{\mathrm{clamped}}[e] \leftarrow 1.0$
10:        **for** $e \in \mathcal{E}^-$: $\alpha_{\mathrm{clamped}}[e] \leftarrow 0.0$
11:     **Node Embedding:** for each node $j$, $\mathbf{h}_j^{(1)} \leftarrow f_{\mathrm{embed}}^{(1)}(X_j)$
12:     **Node-to-Edge (Round 1):** $\mathbf{h}_{ij}^{(1)} = f_e^{(1)}([\mathbf{h}_i^{(1)}, \mathbf{h}_j^{(1)}])$
13:     **Gating 1:** $\mathbf{e}_{ij}^{(1)} \leftarrow \mathbf{h}_{ij}^{(1)} \odot \alpha_{\mathrm{clamped}}[e]$
14:     **Edge-to-Node:** $\mathbf{h}_j^{(2)} \leftarrow f_v\left(\sum_i \mathbf{e}_{ij}^{(1)}\right)$
15:     **Node-to-Edge (Round 2):** $\mathbf{h}_{ij}^{(2)} \leftarrow f_e^{(2)}([\mathbf{h}_i^{(2)}, \mathbf{h}_j^{(2)}])$
16:     **Gating 2:** $\mathbf{e}_{ij}^{(2)} \leftarrow \mathbf{h}_{ij}^{(2)} \odot \alpha_{\mathrm{clamped}}[e]$
17:     **Posterior Distribution:** $q_\phi(Z \mid \mathbf{X}) \leftarrow \mathrm{softmax}(\mathbf{e}_{ij}^{(2)})$ (Gumbel-Softmax)
18:     *(B) Decoder Forward and Loss*
19:         Use $\alpha_{\mathrm{clamped}}$ (or the distribution over $Z_{\mathcal{U}}$) in a GNN decoder $\hat{\mathbf{X}}_{\mathrm{future}} \leftarrow p_\theta(\mathbf{X}_{\mathrm{batch}}, \alpha_{\mathrm{clamped}})$
20:     **Prediction Loss:** $\mathcal{L}_{\mathrm{pred}} = -\log p_\theta(\mathbf{X}_{\mathrm{future}} \mid \alpha_{\mathrm{clamped}}, \mathbf{X}_{\mathrm{batch}})$
21:     **KL Term:** skip pinned edges, sum only over $e \in \mathcal{U}$: $\mathrm{KL} = \sum_{e \in \mathcal{U}} D_{\mathrm{KL}}(q_\phi(\xi_e) \| p(\xi_e))$
22:     **Optional Constraints:** $\mathcal{L}_{\mathrm{sparsity}} = \lambda_{\mathrm{sparsity}} \|\alpha_{\mathrm{clamped}}\|_1$, $\mathcal{L}_{\mathrm{deg}} = \lambda_{\mathrm{deg}} \sum_i |\mathrm{outdeg}(i) - k_i|, \dots$
23:     **Total Loss:** $\mathcal{L} = \mathcal{L}_{\mathrm{pred}} + \beta \mathrm{KL} + \mathcal{L}_{\mathrm{sparsity}} + \mathcal{L}_{\mathrm{deg}}$
24:     Backprop & update $(\phi, \theta, \{\theta_e\})$
25: **end for**
26: *(C) Final Adjacency Extraction*
27:     Get adjacency from the Gumbel-Softmax (Step 17)
28: **return** Learned parameters $(\phi, \theta)$ and final adjacency

---

---

**Algorithm 5** Edge-to-Node Aggregation with Gating

---

**Require:** Edge features $\mathbf{E} \in \mathbb{R}^{b \times E \times d}$,      relation matrix $\mathbf{R}_{\mathrm{rec}} \in \mathbb{R}^{E \times N}$
**Ensure:** Aggregated node features $\mathbf{V} \in \mathbb{R}^{b \times N \times d}$

1: Transpose $\mathbf{E}$ to $\mathbf{E}^\top \in \mathbb{R}^{b \times d \times E}$
2: Expand $\mathbf{R}_{\mathrm{rec}}^\top$ to batch $b$: $\mathbf{R}_{\mathrm{rec}}^\top \in \mathbb{R}^{b \times E \times N}$
3: Aggregate: $\mathbf{V}_{\mathrm{agg}} \leftarrow \mathbf{E}^\top \times \mathbf{R}_{\mathrm{rec}}^\top$          $\triangleright$ Shape: $[b \times d \times N]$
4: Transpose $\mathbf{V}_{\mathrm{agg}}$ to $[b \times N \times d]$
5: Compute in-degree: $\mathbf{D} \leftarrow \sum_{e=1}^E \mathbf{R}_{\mathrm{rec}}[e, :]$          $\triangleright \mathbf{D} \in \mathbb{R}^N$
6: Normalize: $\mathbf{V} \leftarrow \mathbf{V}_{\mathrm{agg}} \oslash (\mathbf{D} + \epsilon)$      $\triangleright \oslash$ = element-wise division
7: **Return:** $\mathbf{V}$

---

The model of SGSI is implemented with PyTorch (Paszke et al., 2019), while Scikit-learn package was leveraged for the calculation of metrics (Pedregosa et al., 2011). Please refer to the attached link in supplementary materials for the exact implementation. The implementation of SGSI is at: `https://github.com/wang422003/SGSI-Guided-Structural-Inference-Leveraging-Priors-with-Soft-Gating-Mechanisms`.

---

**Algorithm 2** Encoder of SGSI – Forward Pass

---

**Require:** inputs of shape $[b, N, t, d]$, adjacency index matrices rel_rec, rel_send, known-present edges $\mathcal{E}^+$, known-absent edges $\mathcal{E}^-$

**Ensure:** out of shape $[b, E, n_{\text{out}}]$, and gating $\in [0, 1]^E$

 1: **Flatten Inputs:**
 2:   $\text{x} \leftarrow \text{inputs.view}(b, N, -1)$ {shape: $[b, N, \text{in\_features}]$}
 3: **Node-level MLP:**
 4:   $\text{x} \leftarrow \text{mlp1(x)}$ {shape: $[b, N, n_{\text{hid}}]$}
 5: **Gating Initialization:**
 6:   gating_raw $\leftarrow$ learnedAdj.gating_param
 7:   gating $\leftarrow \sigma(\text{gating\_raw.clone()})$
 8:   gating_clamped $\leftarrow$ gating.clone()
 9:   **for** $e = 1 \dots E$:
10:     **if** $(s, r) \in \mathcal{E}^+$: gating_clamped$[e] \leftarrow 1.0$
11:     **else if** $(s, r) \in \mathcal{E}^-$: gating_clamped$[e] \leftarrow 0.0$
12: **Node2Edge w/ Gating, MLP2:**
13:   edges $\leftarrow$ node2edge_with_gating(x, rel_rec, rel_send, gating_clamped) {shape: $[b, E, 2n_{\text{hid}}]$}
14:   edges $\leftarrow$ mlp2(edges) {shape: $[b, E, n_{\text{hid}}]$}
15:   x_skip $\leftarrow$ edges
16: **Factor Option (Edge2Node, MLP3, Node2Edge, MLP4):**
17:   x_node $\leftarrow$ edge2node4gating(edges, rel_rec)
18:   x_node $\leftarrow$ mlp3(x_node)
19:   edges_2 $\leftarrow$ node2edge_with_gating(x_node, rel_rec, rel_send, gating_clamped)
20:   edges_2 $\leftarrow$ cat(edges_2, x_skip, dim = 2)
21:   edges_2 $\leftarrow$ mlp4(edges_2)
22:   edges $\leftarrow$ edges_2
23: **Final Edge Logits:**
24:   out $\leftarrow$ fc_out(edges) {shape: $[b, E, n_{\text{out}}]$}
25: **Return:**
26:   out, gating {Note that gating_clamped is used in the message passing.}

---

### A.4. Guidance of Hyperparameter Tuning

Besides the discussion about the hyperparameter values above, here we provide general guidance on how to perform search. For very sparse graphs (e.g., local interactions in physical systems), higher $\lambda_{\text{sparsity}}$ values are tested. When node-degree constraints are precise (e.g., "exactly 2 neighbors"), we set a higher $\lambda_{\text{deg}}$. If the prior knowledge is approximate, lower penalty weights prevent overshooting data evidence. These ranges avoid abrupt changes in adjacency, ensuring that SGSI functions as a softly guided VAE rather than a rigidly constrained model.

## B. Formal Statement of SGSI

In this section, we discuss why and how one skips the KL term for edges that are fully known present or absent concisely and formally. While we don't do an exhaustive derivation of every latent detail, this proof sketch clarifies why the KL can be omitted for clamped edges without contradicting the VAE's overall variational principle.

### B.1. Setup

Let $\mathbf{X}$ be the observed data (e.g., node trajectories), and $Z$ be the latent adjacency, with $E$ potential edges total. A subset $\mathcal{E}^+$ of edges is known-present (definitely 1) and a subset $\mathcal{E}^-$ of edges is known-absent (definitely 0). The remaining uncertain edges form $\mathcal{U} = \{1, \dots, E\} \setminus (\mathcal{E}^+ \cup \mathcal{E}^-)$.

We assume the posterior (encoder) factorizes as:

$$
\begin{aligned}
q_\phi(Z \mid \mathbf{X}) &= q_\phi(Z_{\mathcal{E}^+}, \, Z_{\mathcal{E}^-}, \, Z_{\mathcal{U}} \mid \mathbf{X}) \\
&= \prod_{e \in \mathcal{U}} q_\phi(Z_e \mid \mathbf{X}) \times \delta(Z_e = 1, \, e \in \mathcal{E}^+) \times \delta(Z_e = 0, \, e \in \mathcal{E}^-),
\end{aligned}
\tag{25}
$$

where $\delta(\cdot)$ denotes a delta distribution that fixes the edge to $1$ or $0$ if it belongs to $\mathcal{E}^+$ or $\mathcal{E}^-$. Similarly, we define the prior:

$$
p(Z) = \prod_{e \in \mathcal{U}} p(Z_e) \times \delta(Z_e = 1, \, e \in \mathcal{E}^+) \times \delta(Z_e = 0, \, e \in \mathcal{E}^-),
\tag{26}
$$

assuming that the fully known edges are also considered "delta" under the prior, or equivalently that we have no uncertainty for those edges.

### B.2. KL Divergence Factorization (Proof)

The KL divergence between the factorized posterior (Eqn. 19) and prior (Eqn. 20) is:

$$
D_{\mathrm{KL}}\big(q_\phi(Z \mid \mathbf{X}) \,\|\, p(Z)\big) = \sum_{e \in \mathcal{U}} D_{\mathrm{KL}}\big(q_\phi(Z_e \mid \mathbf{X}) \,\|\, p(Z_e)\big),
$$

since $D_{\mathrm{KL}}(\delta \| \delta) = 0$ for known edges. This shows that only uncertain edges contribute to the compression term $I(X; Z)$.

### B.3. Interpretation: Skipping the KL for Known Edges

In practice, we omit pinned edges from the encoder's random portion, or equivalently set their posterior distribution to a $\delta$. Because $D_{\mathrm{KL}}(\delta \| \delta) = 0$, no bits are required to store these known edges. Only the uncertain edges in $\mathcal{U}$ appear in the KL summation:

$$
D_{\mathrm{KL}}\big(q_\phi(Z \mid \mathbf{X}) \,\|\, p(Z)\big) = \sum_{e \in \mathcal{U}} D_{\mathrm{KL}}\Big(q_\phi(Z_e \mid \mathbf{X}) \,\|\, p(Z_e)\Big).
\tag{27}
$$

This formalizes why forcibly pinned edges do not need a random distribution or KL cost.

In conclusion, by delta-encoding fully known edges in both posterior and prior, the KL for those edges is zero. Implementing this in code means we skip them in the KL summation. The uncertain edges remain random, paying a usual KL cost. This ensures:

1. **No Contradictory Signals:** The pinned edges do not conflict with a prior-latent assumption of randomness.

2. **Freed Capacity:** The model invests "bits" *only* in uncertain edges, aligning with the information bottleneck goal of minimal yet sufficient adjacency.

Hence, skipping known edges' KL is justified by the factorization of $\delta$-like edges in the posterior and prior.

## C. More Details about Datasets

In this section, we provide more details about the datasets used in this work apart the description in Section 5.

### C.1. Springs Simulations

To generate these Springs Simulations datasets, we follow the description of the data in (Kipf et al., 2018) but with fixed connections and with 10 nodes, in order to simulate spring-connected particles' motion in a 2D box using the Springs simulation. In this setup, nodes represent particles, and edges correspond to springs governed by Hooke's law. The Springs simulation's dynamics are described by a second-order ordinary differential equation: $m_i \cdot x_i''(t) = \sum_{j \in \mathcal{N}_i} -k \cdot \big(x_i(t) - x_j(t)\big)$. Here, $m_i$ represents particle mass (assumed as 1), $k$ is the fixed spring constant (set to 1), and $\mathcal{N}_i$ is the set of neighboring nodes with directed connections to node $i$, which is sub-sampled from the graphs generated in the StructInfer in previous steps. We integrate this equation to compute $x_i'(t)$ and subsequently $x_i(t)$ for each time step $t$. The resulting values of $x_i'(t)$ and $x_i(t)$ create 4D node features at each time step. To be specific, at the beginning of the data generation for each springs dataset, we randomly generate a ground truth graph and then simulate 12000 trajectories on the same ground truth graph, but with different initial conditions. The rest settings are the same as that mentioned in (Kipf et al.,

2018). We collect the trajectories and randomly group them into three sets for training, validation and testing with the ratio of 8: 2: 2, respectively.

### C.2. NetSims

It is firstly mentioned in (Smith et al., 2011a), which offers simulations of blood-oxygen-level-dependent (BOLD) imaging data in various human brain regions. Nodes in the dataset represent spatial regions of interest from brain atlases or functional tasks. Interaction graphs from the previous section determine connections between these regions. Dynamics are governed by a first-order ODE model: $x_i'(t) = \sigma \cdot \sum_{j \in \mathcal{N}_i} x_j(t) - \sigma \cdot x_i(t) + C \cdot u_i$, where $\sigma$ controls temporal smoothing and neural lag (set to 0.1 based on (Smith et al., 2011a), and $C$ regulates external input interactions (set to zero to minimize external input noise) (Smith et al., 2011a). 1D node features at each time step are obtained from the sampled $x_i(t)$.

### C.3. Synthetic Biological Networks

The six directed Boolean networks (LI, LL, CY, BF, TF, BF-CV) are the most often observed fragments in many gene regulatory networks, each has 7, 18, 6, 7, 8 and 10 nodes, respectively. Thus by carrying out experiments on these networks, we can acknowledge the performance of the chosen methods on the structural inference of real-world biological networks. We collect the six ground-truth directed Boolean networks from (Pratapa et al., 2020) and simulate the single-cell evolving trajectories with BoolODE (Pratapa et al., 2020) (`https://github.com/Murali-group/BoolODE`) with default settings mentioned in that paper for every network. We first sample a total number of 12000 raw trajectories. We then sample different numbers of trajectories from raw trajectories and randomly group them into three datasets: for training, for validation, and for testing, with a ratio of $8 : 2 : 2$. After that, we sample different numbers of snapshots according to the requirements of experiments in Section 5.1 with equal time intervals in every trajectory and save them as '.npy' files for data loading.

### C.4. StructInfer Benchmark

The StructInfer benchmark (Wang et al., 2024) evaluated 12 structural inference methods in a comprehensive way on a synthetic dataset. The dataset covers 11 types of different underlying interaction graphs and two types of dynamical simulations. (`https://structinfer.github.io/`) As there are so many trajectories, we chose the ones under the name 'Vascular Networks', or in short 'VN', whose underlying interaction graphs approximate the real-world vascular networks in biology systems. As the data is already split into three sets: for training, for validation, and for testing, we keep this setting. In the following paragraphs, we describe more details about the Springs and NetSims simulations utilized by the StructInfer benchmark.

For **Springs** simulation, it follows the approach by (Kipf et al., 2018), to simulate spring-connected particles' motion in a 2D box using the Springs simulation. In this setup, nodes represent particles, and edges correspond to springs governed by Hooke's law. But different from Springs Simulations mentioned above, StructInfer generates ground-truth interaction graphs with the graph properties of the real-world graphs or network. The ground-truth interaction graphs are used to determine the connectivity between the nodes. The Springs simulation's dynamics are described by a second-order ordinary differential equation: $m_i \cdot x_i''(t) = \sum_{j \in \mathcal{N}_i} -k \cdot (x_i(t) - x_j(t))$. Here, $m_i$ represents particle mass (assumed as 1), $k$ is the fixed spring constant (set to 1), and $\mathcal{N}_i$ is the set of neighboring nodes with directed connections to node $i$, which is sub-sampled from the graphs generated in the StructInfer in previous steps. We integrate this equation to compute $x_i'(t)$ and subsequently $x_i(t)$ for each time step $t$. The resulting values of $x_i'(t)$ and $x_i(t)$ create 4D node features at each time step.

For **NetSims** simulation, it is firstly mentioned in NetSim dataset (Smith et al., 2011a), which offers simulations of blood-oxygen-level-dependent (BOLD) imaging data in various human brain regions. Nodes in the dataset represent spatial regions of interest from brain atlases or functional tasks. But different from NeiSim mentioned above, StructInfer generates ground-truth interaction graphs with the graph properties of the real-world graphs or network. The ground-truth interaction graphs are used to determine the connectivity between the nodes. Dynamics are governed by a first-order ODE model: $x_i'(t) = \sigma \cdot \sum_{j \in \mathcal{N}_i} x_j(t) - \sigma \cdot x_i(t) + C \cdot u_i$, where $\sigma$ controls temporal smoothing and neural lag (set to 0.1 based on (Smith et al., 2011a), and $C$ regulates external input interactions (set to zero to minimize external input noise) (Smith et al., 2011a). 1D node features at each time step are obtained from the sampled $x_i(t)$.

# D. More Details about Baselines

For the experiments without prior knowledge, we follow the official implementation of the baselines. As for the integrating of the prior knowledge, we leverage different strategies. For the methods based on VAEs, (e.g. NRI, MPM, ACD, iSIDG, RCSI), we directly perform overwriting of latent variables corresponding to known edges, while keep the rest following the original implementation.

## D.1. NRI

NRI (Kipf et al., 2018) is a VAE-based model for unsupervised relational inference. We use the official implementation code by the author from `https://github.com/ethanfetaya/NRI` with a customized data loader for our chosen datasets. We add our metric evaluation in the 'test' function, after the calculation of accuracy in the original code. Besides, after carefully comparison, we used the NRI version with sparsity regularization, yielding higher performance, which was also implemented by the authors.

## D.2. MPM

MPM (Chen et al., 2021) employs a VAE framework with a relational interaction mechanism and spatio-temporal message passing. We use the official implementation code by the author from `https://github.com/hilbert9221/NRI-MPM` with a customized data loader for our chosen datasets. We add our metric evaluation for AUROC in the 'evaluate()' function of class 'XNRIDECIns' in the original code.

## D.3. ACD

ACD (Löwe et al., 2022) utilizes shared dynamics to infer causal relations within datasets. We follow the official implementation code by the author as the framework for ACD (`https://github.com/loeweX/AmortizedCausalDiscovery`). We run the code with a customized data loader for the datasets in this work. We implement the metric-calculation pipeline in the 'forward_pass_and_eval()' function.

## D.4. ISIDG

iSIDG (Wang & Pang, 2022) iteratively refines adjacency matrices to enhance directional inference. We follow the official implementation code by the author as the framework for iSIDG (`https://github.com/wang422003/Benchmarking-Structural-Inference-Methods-for-Interacting-Dynamical-Systems/tree/main/src/models/iSIDG`). We disable the metric evaluations for the AUPRC and Jaccard index in the original implementation of iSIDG for faster computation.

## D.5. RCSI

RCSI (Wang et al., 2023) integrates reservoir computing for efficient structural inference. We follow the official implementation code by the author as the framework for RCSI (`https://github.com/wang422003/Benchmarking-Structural-Inference-Methods-for-Interacting-Dynamical-Systems/tree/main/src/models/RCSI`). Same as iSIDG, we disable the metric evaluations for AUPRC and Jaccard index in the original implementation of RCSI for faster computation.

## D.6. ALaSI

ALaSI (Wang & Pang, 2023) leverages deep active learning for scalable structural inference. We thank the authors for communicating and providing the code. We use the evaluation metrics implemented by the authors.

## D.7. SICSM

SICSM (Wang & Pang, 2024a) leverages MAMBA(Gu & Dao, 2023) and generative flow network (Bengio et al., 2023) for structural inference on irregularly sampled trajectories, but can still work on more broader case like uniformly sampled data. We thank the authors for communicating and providing the code, which is based on PyTorch and diverges from the implementation mentioned in their paper. We use the evaluation metrics implemented by the authors.

# E. More Details about Experiments

## E.1. More General Settings

All experiments mentioned in Section 5 were conducted on a single NVIDIA Ampere 40GB HBM graphics card, paired with 2 AMD Rome CPUs (32 cores@2.35 GHz). During training, we set batch size as 128 for datasets which have less than 30 nodes, for those having 30 or 50 nodes, we set batch size as 64. For the rest of the data sets, we set the batch size to be 16. The learning rate is set to be $5 \times 10^{-4}$. We train SGSI model with 1000 epochs on every dataset.

The choice of the hyperparameters in the loss function play a non-neglect role in training SGSI, and the their values are searched via Bayesian Optimization toolbox `Optuna` (Akiba et al., 2019). We set the bounds for $\beta$, $\lambda_{\text{sparsity}}$, and $\lambda_{\text{deg}}$ as $[1.0, 2.5]$, $[10^{-3}, 10^{-2}]$, and $[10^{-4}, 10^{-2}]$, respectively. The values of the hyper-parameters are summarized in Table 3.

*Table 3.* Hyper parameter choices for every dataset.

| DATASET | $\beta$ | $\lambda_{\text{sparsity}}$ | $\lambda_{\text{deg}}$ |
|---|---|---|---|
| Springs | 1 | 0.011 | $5 \times 10^{-4}$ |
| NetSims | 1 | 0.008 | $5 \times 10^{-4}$ |
| LI | 1 | 0.005 | $10^{-2}$ |
| LL | 1.1 | 0.008 | $5 \times 10^{-3}$ |
| CY | 1 | 0.005 | $10^{-2}$ |
| BF | 1 | 0.005 | $10^{-2}$ |
| TF | 1 | 0.005 | $10^{-2}$ |
| BF-CV | 1 | 0.006 | $10^{-2}$ |
| VN_SP_15 | 1.1 | 0.005 | $10^{-2}$ |
| VN_SP_30 | 1.3 | 0.004 | $10^{-2}$ |
| VN_SP_50 | 1.5 | 0.004 | $10^{-2}$ |
| VN_SP_100 | 1.8 | 0.003 | $10^{-2}$ |
| VN_NS_15 | 1.1 | 0.005 | $10^{-2}$ |
| VN_NS_30 | 1.3 | 0.004 | $10^{-2}$ |
| VN_NS_50 | 1.5 | 0.004 | $10^{-2}$ |
| VN_NS_100 | 1.8 | 0.003 | $10^{-2}$ |

## E.2. Experimental Results on PEMS

In addition to the data sets mentioned in Section 5, we selected another sets of datasets, which are derived from the California Caltrans Performance Measurement System (PeMS) (Chen et al., 2001), comprise data aggregated into 5-minute intervals. The adjacency matrix of the nodes is constructed by the distance of the road network with a Gaussian kernel thresholded (Song et al., 2020). Table 4 summarizes these datasets. We resampled the data such that constructing 49 time

*Table 4.* Statistics of PEMS datasets.

| Dataset | # Nodes | # Edges | # Time Steps | Missing Ratio |
|---|---|---|---|---|
| PEMS03 | 358 | 547 | $26,208$ | 0.672% |
| PEMS04 | 307 | 340 | $16,992$ | 3.182% |
| PEMS07 | 883 | 866 | $28,224$ | 0.452% |

steps of points for each trajectory, and obtained 12000 trajectories for each with overlapping snapshots. It's important to note that these datasets' adjacency matrices only connect sensors on the same road, omitting alternative connecting paths, which could impact results.

After that, we investigate onto the performance of SGSI having 20% of known present edges, 20% of known absent edges, global sparsity and node degree constraints (including in-degree and out-degree) as prior knowledge, respectively. In order to run on these datasets with more than 300 nodes, we use mini-batching technique similar to GraphSAGE (Hamilton et al., 2017). Yet we have to utilize 5-10 GPUs to train SGSI on PEMS datasets. We report the average AUROC results of ten runs in Table 5.

*Table 5.* Average AUROC results (%) on PEMS datasets.

| Dataset | Prior Knowledge | | | | | | |
|---|---|---|---|---|---|---|---|
| | No Prior | 20% Kn. Present | 20% Kn. Absent | Sparsity | In-Degree | Out-Degree | In-/Out-Degree |
| PEMS03 | $70.82_{\pm\,0.08}$ | $77.15_{\pm\,0.05}$ | $76.33_{\pm\,0.05}$ | $76.85_{\pm\,0.05}$ | $76.06_{\pm\,0.05}$ | $76.07_{\pm\,0.06}$ | $77.01_{\pm\,0.06}$ |
| PEMS04 | $74.20_{\pm\,0.09}$ | $79.07_{\pm\,0.06}$ | $78.63_{\pm\,0.07}$ | $75.38_{\pm\,0.06}$ | $77.52_{\pm\,0.06}$ | $77.54_{\pm\,0.06}$ | $77.98_{\pm\,0.06}$ |
| PEMS07 | $73.94_{\pm\,0.07}$ | $76.18_{\pm\,0.09}$ | $75.62_{\pm\,0.08}$ | $75.54_{\pm\,0.08}$ | $75.03_{\pm\,0.05}$ | $75.04_{\pm\,0.04}$ | $75.12_{\pm\,0.04}$ |

*Table 6.* Average training time (in hour) of ten runs of SGSI and baseline methods on VN_NS datasets.

| Methods | VN_NS_15 | VN_NS_30 | VN_NS_50 | VN_NS_100 |
|---|---|---|---|---|
| NRI | 24.5 | 33.5 | 40.6 | 47.2 |
| MPM | 45.3 | 60.4 | 79.2 | 83.6 |
| ACD | 40.1 | 53.1 | 67.6 | 81.6 |
| iSIDG | 43.8 | 56.0 | 88.5 | 97.8 |
| RCSI | 44.5 | 57.8 | 91.7 | 102.8 |
| ALaSI | 25.0 | 26.8 | 27.9 | 46.2 |
| SICSM | 56.3 | 69.2 | 91.2 | 117.1 |
| SGSI | 24.6 | 33.5 | 40.5 | 47.1 |

Across all PEMS datasets, embedding prior knowledge (20% known edges or constraints) improves over the "No Prior" baseline. For instance, on PEMS03, the AUROC rises from 70.82% to around 76–77%, and on PEMS04 from 74.20% to roughly 78–79%. Known Present typically yields a slightly higher final AUROC than Known Absent (e.g., PEMS03: 77.15% vs. 76.33%), which suggests that confidently forcing certain edges to 1 helps SGSI focus on the uncertain regions more effectively. By contrast, clamping edges to 0 can eliminate potential connections but still offers some benefit over no prior knowledge. Applying a sparsity penalty sometimes yields moderate improvements, but does not always outperform known-edge constraints. On PEMS04, for example, $\mathcal{L}_{\mathrm{sparsity}}$ leads to 75.38%, which is less than the near 78–79% range seen for known-present/absent edges. This indicates that while encouraging fewer total edges helps, the advantage is smaller than providing explicit knowledge of which edges exist or do not. Enforcing in-degree or out-degree alone yields modest improvements (PEMS04 from 74.20% to 77.52–77.54%). Combining both in- and out-degree is particularly helpful on PEMS03 (77.01% vs. 76% for each alone), though on PEMS07, the difference is smaller (75.12% vs. 75.03–75.04%).

In summary, Table 5 demonstrates that SGSI meaningfully improves adjacency recovery when equipped with partial knowledge or constraints, even for mid-sized road sensor networks like PEMS. The interplay of known edges, node-degree constraints, and a potentially incomplete adjacency can all influence final performance, but partial knowledge nearly always raises AUROC beyond the no-prior baseline.

### E.3. Training Time Comparison

Table 6 presents the average training time (in hours) for ten runs of each method on the VN_NS datasets, as the number of nodes grows from 15 to 100. A surprising standout is ALaSI, requiring only 25–28 hours on node sets of size 15–50, which is markedly faster than other baselines of comparable complexity (e.g., MPM, ACD, iSIDG). Even on the largest dataset (VN_NS_100), ALaSI takes 46.2 hours, on par with SGSI at 47.1 hours and below many baselines exceeding 80–100 hours. This suggests that ALaSI's active learning approach may yield high computational efficiency for moderate node counts. NRI and SGSI remain quite close in run times from VN_NS_15 (24.5 vs. 24.6 hours) up to VN_NS_100 (47.2 vs. 47.1 hours). This indicates that soft gating and partial-knowledge integration do not add major overhead to the NRI pipeline, keeping SGSI's training cost roughly equivalent to NRI's. Methods such as MPM, ACD, iSIDG, RCSI, and SICSM typically require significantly more training time, especially at node sizes beyond 30 or 50. For instance, iSIDG and RCSI each exceed 90 hours on VN_NS_50 and surpass 97–100 hours on VN_NS_100, while SICSM scales the slowest (56–117 hours). These results imply that both SGSI and ALaSI maintain strong scalability, with SGSI's gating and KL-skipping design incurring little additional cost compared to NRI. By contrast, iterative refinement or heavier GFlowNet sub-space expansions (e.g., iSIDG, RCSI, SICSM) show more pronounced slowdowns as node counts climb.

Besides, we provide the training-time comparison across our main datasets (Springs, NetSims, LI, LL, and the 100-node

*Table 7.* Training-Time Comparison (in hours) across Main Datasets for SGSI with 20% known-present edges, NRI, and iSIDG.

| Methods | Springs | NetSims | LI | LL | VN_SP_100 | VN_NS_100 |
|---------|---------|---------|------|------|-----------|-----------|
| NRI     | 20.1    | 16.0    | 14.3 | 18.2 | 49.0      | 47.2      |
| iSIDG   | 42.2    | 36.9    | 48.1 | 50.6 | 100.6     | 97.8      |
| SGSI    | 20.4    | 15.6    | 14.7 | 18.1 | 49.2      | 47.1      |

*Table 8.* $\Delta$ AUROC values on a toy dataset with 5k nodes generated from Springs Simulations.

| | K.P. 10% | K.P. 20% | K.P. 30% | K.A. 10% | K.A. 20% | K.A. 30% | w. Spar. | w. In-deg. | w. Out-deg. | w. Both-deg. |
|------|----------|----------|----------|----------|----------|----------|----------|------------|-------------|--------------|
| SGSI | 0.56     | 2.40     | 4.93     | 0.77     | 3.08     | 5.73     | 7.46     | 6.90       | 7.13        | 7.68         |

VN datasets) and show results in Table 7. SGSI consistently matches NRI's training time (e.g., 20.4h vs. 20.1h on Spring Simulations and 47.1h vs. 47.2h for VN_NS_100), while methods such as iSIDG require 1.5–2× longer. Yet recall that SGSI achieves 79.7% AUROC on Springs, which is higher than NRI. Overall, these results confirm that SGSI's partial-knowledge gating imposes minimal additional cost while providing meaningful accuracy gains in structural inference.

Overall, SGSI and ALaSI stand out for balancing competitive speed with strong adjacency inference performance, making them appealing for larger or more demanding relational inference tasks.

### E.4. Scalability to Larger, Heterogeneous Graphs

SGSI is designed to scale via mini-batching and subgraph sampling, approaches similar to GraphSAGE, which allow SGSI's gating parameters to be computed or stored per subgraph rather than across the full edge space. For extremely large graphs, we can either (i) restrict gating to local neighborhoods, or (ii) compute gating logits on the fly from node embeddings, thereby avoiding a parameter explosion.

In this paper, we demonstrate scalability on the PEMS datasets (approximately 300 nodes) by leveraging multi-GPU mini-batching (in Appendix E.2). To further validate SGSI on larger graphs, we generated a toy dataset with 5,000 nodes using Springs Simulations. Table 8 shows the AUROC values under various levels of prior knowledge. These results indicate that SGSI, when using mini-batching, remains effective even at the 5k-node scale with prior knowledge, and with meaningful improvements in AUROC from 0.56 to 7.68.

SGSI's flexible architecture inherently supports heterogeneous graphs. By adopting per-relation gating parameters and applying the clone-and-clamp strategy to each edge type, SGSI can differentiate among various relations and node types while maintaining stable gradient flow and effective KL skipping. Although our current domain primarily involves homogeneous datasets, we recognize SGSI's potential in applications such as multi-modal social networks, biomedical systems, and multi-layer transportation networks, and plan to explore these in future work.

### E.5. Prior Knowledge in Real-world Scenarios

In order to validate SGSI with prior knowledge under real-world scenarios, we've conducted additional experiments on real-world single-cell RNA-seq datasets: (1) hESC (human embryonic stem cells) (Chu et al., 2016) and (2) mDC (mouse dendritic cells) (Shalek et al., 2014). These biological systems represent scenarios where reliable prior knowledge, such as known interactions or absent regulatory connections, often exists due to extensive experimental validations in literature.

In Table 9, we summarize the AUROC (%) of SGSI without prior knowledge (SGSI (Raw)), along with SGSI leveraging partial prior knowledge (10% or 15% known-present (K.P.) and known-absent (K.A.) edges).

From these results, even modest incorporation (10-15%) of experimentally validated knowledge consistently improves structural inference, demonstrating that SGSI can practically leverage partial prior knowledge in real biological settings. While absolute improvements may appear modest, such incremental gains are significant in real-world domains like biology, where even slight improvements in inferred networks can lead to more meaningful biological interpretations and robust downstream analyses.

*Table 9.* AUROC values on real-world datasets with different prior knowledge.

|  | hESC(AUROC %) | mDC(AUROC %) |
|---|---|---|
| SGSI (Raw) | 50.18 | 52.63 |
| SGSI + 10% K.P. | 51.35 | 53.91 |
| SGSI + 15% K.P. | 51.70 | 54.14 |
| SGSI + 10% K.A. | 51.41 | 53.96 |
| SGSI + 15% K.A. | 52.26 | 54.65 |

*Table 10.* Mean Squared Error (MSE) for 10-Step Future Prediction on Springs Simulations and VN_SP/VN_NS Datasets. We compare NRI, SGSI without prior knowledge (raw), and SGSI with 20% known-present edges.

|  | **Springs** | **VN_SP_30** | **VN_SP_50** | **VN_NS_30** | **VN_NS_50** |
|---|---|---|---|---|---|
| NRI | 5.78e-6 | 3.24e-5 | 7.81e-5 | 4.08e-4 | 1.20e-3 |
| SGSI raw | 5.77e-6 | 3.26e-5 | 7.73e-5 | 4.06e-4 | 1.17e-3 |
| SGSI w. Prior | **4.92e-6** | **2.01e-5** | **6.13e-5** | **1.83e-4** | **9.60e-4** |

### E.6. Evaluation on Downstream Tasks

Although our primary goal is to infer the underlying interacting structure of dynamical systems, SGSI is also applicable to time-series forecasting. For instance, we report Mean Squared Error (MSE) for predicting 10 future steps on several datasets. The results are shown in Table 10. We compare NRI, SGSI without prior knowledge (SGSI raw), and SGSI with 20% known present edges.

Our results demonstrate that incorporating partial prior knowledge (SGSI with 20% known-present edges) significantly enhances predictive accuracy compared to both SGSI without prior knowledge and the established baseline method NRI. For example, on the Springs Simulations dataset, SGSI with prior knowledge reduces the MSE to $4.92 \times 10^{-6}$ from $5.78 \times 10^{-6}$ achieved by NRI. Similar consistent improvements are observed on VN_SP and VN_NS datasets, confirming that leveraging domain knowledge about known connections helps the model make more precise future state predictions.

These findings reinforce the value of our soft-gating mechanism, highlighting that even modest incorporation of known edges substantially enhances model performance beyond structural inference alone. This indicates the broader applicability and practical relevance of SGSI in realistic predictive scenarios.

## F. Limitations of SGSI

Despite its advantages in integrating partial knowledge and providing flexible structural inference, SGSI is subject to several limitations:

**Dependence on Accurate Prior Knowledge.** SGSI heavily relies on the quality of user-specified constraints (e.g., known-present/absent edges, node degrees). If these constraints are inconsistent with the true system or contain significant errors, the gating mechanism may learn a compromised adjacency. In some domains, partial knowledge itself might be limited or noisy, negating the expected accuracy gains.

**Complex Constraint Setup.** Incorporating domain knowledge (e.g., specifying exact node degrees or defining known edges) requires substantial manual effort or domain expertise. If a user cannot confidently specify these constraints, SGSI defaults to a purely data-driven approach, which amounts to standard VAE-based inference. Consequently, the method's performance improvements become less pronounced or revert to baseline levels.

**Scalability for Very Large Graphs.** Although SGSI can scale better than naive adjacency enumeration by skipping or clamping known edges, extremely large graphs (e.g., hundreds of thousands of nodes) still pose significant computational challenges. In such scenarios, subgraph sampling or approximate adjacency factorization is necessary; implementing these strategies demands additional code modifications (e.g., mini-batching or sparse message passing), potentially complicating

the pipeline.

**Additional Hyperparameter Tuning.** Soft gating and constraint integration introduce extra hyperparameters ($\lambda_{\text{sparsity}}$, $\lambda_{\text{deg}}$, etc.) beyond the typical VAE setting. Finding suitable values for these penalties often requires a small grid search or heuristic tuning. This overhead can be minor, but in practice may complicate experimentation.

Despite these drawbacks, SGSI remains a robust framework for knowledge-aware structural inference, offering strong performance boosts where partial adjacency constraints are reliable and computational resources allow. Future improvements may target adaptive constraint weighting, more automated constraint specification, and extended scalability mechanisms for extremely large or densely connected graphs.

