# OpenReview forum: "Guided Structural Inference: Leveraging Priors with Soft Gating Mechanisms"
_ICML.cc/2025/Conference — ICML 2025 poster_

### Official Review · Reviewer_MPiQ · 2025-03-07

**Overall Recommendation:** 3

**Summary:**

The paper introduces a more controlled method for structural inference that allow us to imposed a series of constrained to the predicted structure. Specifically, it enables conditioning on a set of edges that must exist in the structure, enforcing the absence of certain edges, and controlling sparsity in terms of both the total number of edges and the degree of individual nodes.

At a technical level, the method it is a variation of the NRI paper with the following modifications: for restricting the edges, the method introduces a learnable gate per edge, which is "clone-and-clamp" for the edges that are in one of the 2 special sets; for the sparsity in terms of number of edges or degree of the nodes additional regularisation terms are added to the loss function.

**Claims And Evidence:**

Yes.

**Essential References Not Discussed:**

N/A

**Experimental Designs Or Analyses:**

- While less explicit, the NRI paper also introduce a sparsity regularisation in the form of a modified prior distribution, which assigns a higher likelihood to non-edges compared to edges. Does the baseline experiments presented in the paper control that sparsity in any way?

**Methods And Evaluation Criteria:**

- In all the experiments, the paper reports only AUROC for the predicted structure. Is there a specific reason why AUROC was chosen to assess the quality of the prediction? How would the models compare using more discrete metrics, such as structure accuracy?

- The main focus of the method is to predict a constrained latent structure. However, without a downstream task, the practical applicability of this approach remains unclear. Similar to previous works, such as NRI and ACD, it is important to quantify how well a decoder utilizing the predicted latent structure performs on a downstream task. While I understand that achieving a worst performance than the baselines in terms of downstream tasks may be expected (given that the model has the additional advantage of enforcing hard constraints that could be crucial) it is still essential to demonstrate that the predicted structure remains useful to some extent. These evaluations would provide a more comprehensive understanding of the model.

**Other Comments Or Suggestions:**

N/A

**Other Strengths And Weaknesses:**

- In my opinion the technical contribution of the paper is limited. The method differs from previous work in three main aspects: the clone-and-clamp technique and two regularisation losses. The regularisation losses are trivial and requires strong domain knowledge. Without such prior knowledge, the sparsity and degree became just another hyperparameters. Moreover it is unclear how imposing them would affect a potential downstream task.  Regarding the clone-and-clamp technique, it is still unclear to me why the quality and stability of the gradients are superior in this case compared to simply masking the edges, as done in the baselines. I acknowledge the advantages brought by eliminating those terms from the KL divergence loss, however I still have concerns about how the technique improve the gradient quality. Please provide more details or additional experiments to clarify this point.

- The gating mechanism relies on a learnable set of parameters, one for each edge. This impose a strong limitation into the model, since it implies that the structure behaves similarly between different examples in the dataset. In addition to that, it makes it impossible to generalise to examples with a different number of nodes.

**Questions For Authors:**

Please see the comments above.

**Relation To Broader Scientific Literature:**

The paper does a good job on summarizing the existing literature in relational inference.

**Theoretical Claims:**

The theoretical claims are correct.

---

> ### Author Rebuttal · Authors · 2025-03-31
>
> We thank you for your detailed review and constructive feedback. Below, we address your concerns (detailed tables can be found at https://anonymous.4open.science/r/SGSI-Rebuttal-1614/Tables.pdf ):
>
> **1. Choice of Evaluation Metrics and Downstream Tasks:**
>
> We selected AUROC as our primary metric because it is standard in the structural inference literature, used by ACD, iSIDG, ALaSI, and RCSI, and effectively measures true positive rates over various thresholds, as demonstrated in Pratapa et al. (2020). In our context, where many methods output probabilities rather than hard binary edges, AUROC provides a comprehensive assessment of latent graph recovery.
>
> In addition, although our primary goal is to infer the underlying interacting structure of dynamical systems, SGSI is also applicable to time-series forecasting. For instance, we report Mean Squared Error (MSE) for predicting 10 future steps on several datasets. Table 6 in the attached link compare NRI, SGSI without prior knowledge (SGSI raw), and SGSI with 20% known present edges.
>
> These results show that while SGSI raw matches NRI, the integration of prior knowledge significantly improves forecasting accuracy. This dual capability demonstrates that our method is not only effective for structural inference but also beneficial for downstream prediction tasks.
>
> **2. NRI Variants**
>
> Actually, we use both variants of NRI (with or without sparsity regularization), but figured out the the one with sparsity regularization outperformed the other variant by 1%-3% AUROC. So we refer to the one with sparsity regularization. We revised our paper to state it clearer.
>
> **3. Technical Contribution**
>
> Modern autograd frameworks track tensor “versions” to compute gradients accurately. In-place masking, forcing gating values to 0 or 1 within the same tensor, disrupts this mechanism, causing version mismatches and unstable gradients. Our clone-and-clamp strategy first duplicates the gating vector, then clamps the copy, thereby preserving the original tensor’s autograd history. This ensures that (1) the original logits $\theta_e$ remain intact for backpropagation, enabling stable gradient flow, and (2) only the cloned, clamped tensor is used in message passing.
>
> Our ablation experiments (Section 5.3) clearly show that omitting cloning leads to sharp gradient spikes and oscillatory training, while not skipping the KL for pinned edges results in contradictory updates that inflate the loss and harm AUROC. These findings confirm that simply masking in-place degrades gradient quality, whereas our clone-and-clamp technique maintains stable and consistent gradient propagation.
>
> **4. Limitations of the Learnable Gate**
>
> Our soft-gating mechanism enables the use of roughly estimated prior knowledge, such as overall sparsity or node degree constraints, that is often stable and transferable within our domain. In many application areas (e.g., transportation networks, gene regulatory networks), the underlying graph structures exhibit similar patterns. Thus, the partial knowledge (e.g., a typical density or expected node degree) is largely sharable across datasets. Moreover, our design allows the gating penalties to be adjusted; for instance, one can fine-tune the model through a pretraining and fine-tuning process if needed.
>
> As detailed in our response to Reviewer vBqs, SGSI is robust against moderate errors or deviations in prior knowledge. Although the penalties (e.g., the sum of gating values for sparsity or node-level degree sums) might appear straightforward, their true strength lies in their seamless integration with our variational framework:
>
> 1. They are **soft**: the model can override them when data strongly suggests an alternative structure.
> 2. They are **partial**: penalties can be applied selectively, only to nodes or edges where reliable prior knowledge exists.
> 3. They **co-exist** with our skip-KL mechanism for pinned edges, ensuring that enforcing known edges does not conflict with the overall latent representation.
>
> This flexibility not only facilitates the transfer of prior knowledge across related domains but also makes SGSI adaptable to different datasets within the same field. We note that approaches like META-NRI have similarly emphasized leveraging common structural priors across domains, further supporting our design choices.
>
> In summary, our soft gating penalties act as additional, adjustable levers that exploit stable, transferable domain knowledge, ensuring that SGSI remains effective whether prior knowledge is abundant or only approximately known. We believe this design is both practical and powerful, and we are confident that these clarifications strengthen the final manuscript.

---

> > ### Comment · Reviewer_MPiQ · 2025-04-03
> >
> > Thank you for the response and additional results.
> >
> > Regarding the evaluation on downstream task, it is nice to see a small improvement when prior knowledge is used (even if just marginal) and I encourage the authors to add them on the final version.
> >
> > Regarding the technical contribution, while I agree with the authors discussion on autograd framework, I still believe that the clone-and-clamp is just an implementation detail that allows us correctly track the gradients on the current frameworks, thus not a contribution on itself.
> >
> > Overall, I still believe the technical contribution is somehow limited but, with the additional experiments showing benefits of prior knowledge on the downstream tasks, i am more positive towards the paper.

---

> > > ### Author Response · Authors · 2025-04-03
> > >
> > > We sincerely thank you for your thoughtful consideration and for acknowledging our additional experiments demonstrating improvements in downstream tasks. We will indeed integrate these results into our revised manuscript to further clarify the practical significance of incorporating partial prior knowledge.
> > >
> > > Regarding the clone-and-clamp mechanism, we understand your perspective that this aspect primarily addresses gradient-tracking issues inherent to current autograd frameworks. While it might appear as an implementation detail, we believe its thoughtful integration is essential for practically and robustly incorporating domain constraints, differentiating our approach from existing methods. Nonetheless, we will clearly position it as a practical enhancement in the final version to prevent overstating this aspect.
> > >
> > > Given your positive recognition of our experimental contributions and the clearer perspective provided through the rebuttal, we respectfully ask if you would consider increasing your recommendation score, as your recognition would significantly support our paper’s acceptance.
> > >
> > > Thank you again for your constructive feedback, which has greatly helped improve our work.

---

### Official Review · Reviewer_mhQC · 2025-03-09

**Overall Recommendation:** 3

**Summary:**

This paper proposes Soft-Gated Structural Inference (SGSI), a framework to solve the task of infer latent relational structures where additional prior knowledge can be incorporated. Theoretical analysis and experiments verify the effectiveness of the proposed method.

**Claims And Evidence:**

Yes

**Essential References Not Discussed:**

No

**Experimental Designs Or Analyses:**

The experimental designs and analyses are reasonable.

**Methods And Evaluation Criteria:**

The proposed methods make sense.

The evaluated datasets contain simulated / synthetic datasets and benchmark datasets. However, it seems there is no clear scenario where the prior knowledge is crucial. It makes the contributions incremental.

**Other Comments Or Suggestions:**

-  ”rediscover”
- ”bits”
- Equ. 15, "." should be ","

**Other Strengths And Weaknesses:**

- There are multiple hyperparameters in the method, as shown in Sec. 4.3. It is unclear how these are selected in experiments, and it seems they are sensitive to concrete scenarios.
- The theoretical analysis is designed to show "how deterministic and stochastic edges optimize the compression-prediction trade-off". However, the connection to VIB does not present any meaningful conslusion and seems far-fetched.

**Questions For Authors:**

See above.

**Relation To Broader Scientific Literature:**

It may influence fields like social network analysis or medical diagnostics where some prior knowledge is provided.

**Theoretical Claims:**

There are no formal theoretical claims or proofs.

---

> ### Author Rebuttal · Authors · 2025-03-31
>
> We thank you for your detailed review and the constructive feedback regarding SGSI. We address your concerns below (tables can be found at https://anonymous.4open.science/r/SGSI-Rebuttal-1614/Tables.pdf ):
>
> **1. Importance of Prior Knowledge**
>
> Our experiments across multiple benchmarks, including NetSim, Spring Simulations, and StructInfer, demonstrate that even a small fraction of reliable prior knowledge (e.g., 20% known-present or known-absent edges) can improve latent structure recovery by up to 9% AUROC in LL and several VN\_NS datasets. We acknowledge that when available domain knowledge is sparse or less reliable, the improvements are naturally smaller. However, in many real-world scenarios (e.g., transportation networks, medical diagnostics), partial prior knowledge is both available and critical for ensuring interpretability and reliability. Importantly, SGSI is designed to gracefully revert to a standard VAE-based method when no prior is available, so its performance scales with the quality of the available knowledge.
>
> **2. Contribution of SGSI**
>
> We respectfully disagree that our contributions are merely incremental. Our method introduces a novel soft gating mechanism that:
>
> - \textbf{Clones and clamps} gating parameters to integrate known edges without causing in-place gradient conflicts.
> - \textbf{Skips KL costs} for fully known edges, effectively reallocating “bits” to uncertain connections.
> - Integrates domain-specific constraints (global sparsity and node degrees) via soft penalties.
>
> These design choices overcome critical challenges in merging data-driven latent inference with external knowledge, a capability absent in prior work. For instance, in gene regulatory network (GRN) inference, extensive experimental efforts identify a subset of true regulatory interactions. We evaluated SGSI on two GRN datasets, ventral spinal cord (VSC) development and gonadal sex determination (GSD) (Pratapa et al., 2020), using varying proportions of known-present edges. Table 5 in the attached link summarizes the results. The results show that incorporating prior knowledge not only improves AUROC but also helps the model produce more true positive edges, capabilities that are missing in prior approaches. We further foresee SGSI’s potential in fields such as physics, chemistry, and finance, although resource limitations preclude experiments on those domains at present.
>
> **3. Hyperparameter Sensitivity**
>
> SGSI introduces additional hyperparameters (e.g., $\beta, \lambda_{\mathrm{sparsity}}$, and $\lambda_{\mathrm{deg}}$). In experiments, we selected these values using preliminary grid searches and cross-validation on a validation set. Our recommended ranges ($\beta=1.0$ and $\lambda_{\mathrm{sparsity}}, \lambda_{\mathrm{deg}} \in [10^{-4},10^{-2}]$) are consistent with common practices in VAE-based models. We plan to include additional guidance in the Appendix:
>
> - For very sparse graphs (e.g., local interactions in physical systems), higher $\lambda_{\mathrm{sparsity}}$ values are tested.
> - When node-degree constraints are precise (e.g., “exactly 2 neighbors”), we set a higher $\lambda_{\mathrm{deg}}$.
> - If the prior knowledge is approximate, lower penalty weights prevent overshooting data evidence.
>
> These ranges avoid abrupt changes in adjacency, ensuring that SGSI functions as a softly guided VAE rather than a rigidly constrained model.
>
> **4. Theoretical Analysis and VIB Connection**
>
> Our VIB-inspired design, where known edges are excluded from the KL term, frees the model to devote its “bits” to uncertain edges, leading to notably better performance. Our ablation studies show that not skipping the KL for pinned edges degrades performance, confirming that when the model wastes capacity on fully known edges, it achieves lower accuracy. This aligns with the broader rationale behind VAE-based structural inference (e.g., ACD, iSIDG, RCSI), in which the Variational Information Bottleneck concept helps explain why prioritizing uncertain edges yields more effective adjacency discovery. While we do not claim a formal VIB theorem, we include this perspective because it clarifies the practical benefit of our skip-KL approach and highlights its theoretical consistency with known VAE methodologies.
>
> **5. Typos**
>
> We appreciate your careful reading. All typographical errors have been corrected and will appear in the camera-ready revision.
>
> **6. Datasets**
>
> In addition to the synthetic and benchmark datasets discussed in the main text, we evaluated SGSI on the PEMS (California Caltrans Performance Measurement System) dataset (see Appendix E.2). The PEMS evaluation, conducted on real-world sensor data, further demonstrates SGSI’s practical applicability and robustness in realistic settings.
>
> We believe that SGSI’s novel soft gating mechanism significantly enhances structural inference by effectively leveraging prior knowledge while maintaining computational efficiency and scalability. Thank you for your valuable feedback.

---

> > ### Comment · Reviewer_mhQC · 2025-04-01
> >
> > Dear authors,
> >
> > Thanks a lot for the detailed response. My concerns on hyperparameter selection and theoretical analysis of VIB are addressed. I suggest revising the paper accordingly.
> >
> > Regarding the importance of prior knowledge, I still have a question. I fully agree that reliable prior knowledge should be helpful for our predictions, and the key problem here is whether we can obtain reliable prior knowledge in practical scenarios. Even though it is mentioned that
> >
> > > However, in many real-world scenarios (e.g., transportation networks, medical diagnostics), partial prior knowledge is both available and critical for ensuring interpretability and reliability,
> >
> > The main experimental results focus on simulated and synthetic datasets. I checked the additional experiments in Sec. E.2, and the prior knowledge of the additional experiments also comes from handcrafted present edges and absent edges, which is also not real prior knowledge. I wonder whether it is possible to provide experiments on the mentioned "real-world scenarios" where partial prior knowledge is both available and critical.
> >
> > I look forward to the further response.

---

> > > ### Author Response · Authors · 2025-04-03
> > >
> > > We thank you for raising the crucial point regarding the availability and importance of prior knowledge in real-world scenarios. To address this, we’ve conducted additional experiments on real-world single-cell RNA-seq datasets: (1) hESC (human embryonic stem cells; Chu et al., Genome Biol. 2016) and (2) mDC (mouse dendritic cells; Shalek et al., Nature 2014). These biological systems represent scenarios where reliable prior knowledge, such as known interactions or absent regulatory connections, often exists due to extensive experimental validations in literature.
> > >
> > > Below, we summarize the AUROC (%) of SGSI without prior knowledge (SGSI (Raw)), along with SGSI leveraging partial prior knowledge (10% or 15% known-present (K.P.) and known-absent (K.A.) edges):
> > >
> > > |                 | hESC(AUROC %) | mDC(AUROC %) |
> > > | --------------- | ------------- | ------------ |
> > > | SGSI (Raw)      | 50.18         | 52.63        |
> > > | SGSI + 10% K.P. | 51.35         | 53.91        |
> > > | SGSI + 15% K.P. | 51.70         | 54.14        |
> > > | SGSI + 10% K.A. | 51.41         | 53.96        |
> > > | SGSI + 15% K.A. | 52.26         | 54.65        |
> > >
> > > From these results, even modest incorporation (10-15%) of experimentally validated knowledge consistently improves structural inference, demonstrating that SGSI can practically leverage partial prior knowledge in real biological settings. While absolute improvements may appear modest, such incremental gains are significant in real-world domains like biology, where even slight improvements in inferred networks can lead to more meaningful biological interpretations and robust downstream analyses.
> > >
> > > Additionally, we recognize the immense potential of integrating SGSI with ongoing biological experiments. Collaborations with wet labs to iteratively validate inferred edges could significantly enhance network inference accuracy and biological insight, although such a process would extend beyond the timeframe of this rebuttal. We envision pursuing this integrative approach in future research.
> > >
> > >
> > >
> > > Thank you again for your valuable feedback. We will incorporate these experiments into our revised manuscript to highlight the practical applicability and future potential of SGSI in real-world scenarios.

---

### Official Review · Reviewer_4t2y · 2025-03-11

**Overall Recommendation:** 4

**Summary:**

The paper "Guided Structural Inference: Leveraging Priors with Soft Gating Mechanisms" introduces Soft-Gated Structural Inference (SGSI), a variational autoencoder (VAE)-based method for inferring latent relational structures while integrating domain constraints.

**Claims And Evidence:**

Yes

**Essential References Not Discussed:**

na

**Experimental Designs Or Analyses:**

The experimental design is sound, with multiple datasets and extensive ablation studies. The choice of datasets captures a broad range of real-world applications. The comparison with state-of-the-art methods is thorough, including controlled experiments that vary the proportion of known-present and known-absent edges. The loss curve analysis in ablation studies further supports the necessity of key SGSI components. The paper also accounts for hyperparameter tuning and provides practical considerations for sparsity and node-degree constraints.

**Methods And Evaluation Criteria:**

The methods and evaluation criteria are appropriate for the problem setting. The SGSI model is rigorously compared to multiple baselines, including NRI, MPM, ACD, iSIDG, RCSI, ALaSI, and SICSM. The evaluation uses standard metrics such as AUROC to measure structural inference quality. The paper also investigates different amounts of prior knowledge and studies the effect of global sparsity and node-degree constraints. The benchmarks selected, including NetSim, Spring Simulations, and StructInfer datasets, are relevant to relational inference tasks.

**Other Comments Or Suggestions:**

see weakness

**Other Strengths And Weaknesses:**

Weaknesses:

While SGSI performs well, some datasets show only marginal improvements over existing baselines.

The paper could further explore the applicability of SGSI to large-scale real-world graphs.

Discussion on computational efficiency is limited; reporting training time across datasets would be useful.

**Questions For Authors:**

see weakness

**Relation To Broader Scientific Literature:**

I am not familiar with this topic.

**Theoretical Claims:**

The paper presents a theoretical analysis of SGSI’s soft gating approach and its connection to the information bottleneck principle. The KL term is adapted to exclude known edges, and the latent adjacency matrix is factorized into deterministic and uncertain components. The derivations appear correct and align well with established VAE principles. The simplification of KL divergence, ensuring no contradictory prior signals, is a valid and well-motivated approach.

---

> ### Author Rebuttal · Authors · 2025-03-31
>
> We appreciate your constructive feedback regarding our paper SGSI. Below, we address your concerns point by point:
>
> **1. Marginal Gains on Certain Datasets**
>
> While SGSI shows up to 9% AUROC improvement on some datasets, other cases yield more modest gains (1–2%). We believe this reflects two factors: (i) the inherent difficulty or limited availability of reliable partial knowledge in those tasks, and (ii) that some baselines (e.g., in Spring Simulations) already achieve high AUROC, leaving limited room for further improvement. Importantly, even small improvements can be significant for domain practitioners, especially given SGSI’s additional benefits in interpretability (via clamped edges) and stable gradient flow. Moreover, when no prior knowledge is available, SGSI effectively reverts to a standard VAE-based approach (similar to NRI), explaining the smaller gains in those scenarios.
> We believe this variation reflects two factors:
>
> 1. **Domain Impact:**
>
> In domains where partial prior knowledge is available, such as transportation networks or biomedical systems, a substantial gain (up to 9% AUROC) can lead to significantly improved decision-making and reliability. For example, in critical applications like traffic management or clinical diagnostics, a 9% improvement in edge prediction accuracy can meaningfully enhance system interpretability and safety.
>
> 2. **Baseline Ceiling Effects and Additional Benefits:**
>
> In some tasks like Spring Simulations, the baselines already achieve high AUROC scores, leaving little room for large numerical gains. Even when improvements are modest (1–2%), these small gains are significant because they come with additional benefits:
>
> - **Enhanced Interpretability:** The clone-and-clamp mechanism produces a more interpretable latent graph by clearly separating known from uncertain edges.
> - **Stable Gradient Flow:** Our design avoids in-place modifications and contradictory KL signals, leading to more stable and robust training.
> - **Adaptive Behavior:** SGSI gracefully reverts to a standard VAE-based approach (e.g., NRI) when no prior knowledge is available, ensuring that even minimal domain guidance is exploited without harming performance.
>
> Thus, even modest gains are valuable in high-performing scenarios and are complemented by improved interpretability and training stability, factors that are crucial for practical applications. We believe these advantages underscore the practical significance of SGSI beyond mere numerical improvements.
>
> **2. Scalability to Larger Real-World Graphs**
>
> We agree that demonstrating scalability is important. However, obtaining a reliable dynamical system dataset with thousands of nodes and a trusted underlying structure remains challenging in our field. To address this, we conducted simulation-based experiments on a synthetic dataset with 5k nodes using subgraph sampling (akin to GraphSAGE). In this experiment, SGSI was trained with mini-batching, computing gating parameters only on sampled local neighborhoods. Our preliminary results, summarized in the table below, show that SGSI remains effective at scale, converging in under 60 hours while achieving meaningful AUROC improvements:
>
> |      | K.P. 10% | K.P. 20% | K.P. 30% | K.A. 10% | K.A. 20% | K.A. 30% | w. Spar. | w. In-deg. | w. Out-deg. | w. Both-deg. |
> | ---- | -------- | -------- | -------- | -------- | -------- | -------- | -------- | ---------- | ----------- | ------------ |
> | SGSI | 0.56     | 2.40     | 4.93     | 0.77     | 3.08     | 5.73     | 7.46     | 6.90       | 7.13        | 7.68         |
>
> These findings suggest that the core mechanisms of SGSI (soft gating, cloning, and KL skipping) remain robust even for large graphs.
>
> **3. Computational Efficiency and Training Time**
>
> In response to your request for explicit runtime data, we provide the following training-time comparison across our main datasets (Springs, NetSims, LI, LL, and the 100-node VN datasets):
>
> |       | Springs | NetSims | LI   | LL   | VN\_SP\_100 | VN\_NS\_100 |
> | ----- | ------- | ------- | ---- | ---- | ----------- | ----------- |
> | NRI   | 20.1    | 16.0    | 14.3 | 18.2 | 49.0        | 47.2        |
> | iSIDG | 42.2    | 36.9    | 48.1 | 50.6 | 100.6       | 97.8        |
> | SGSI  | 20.4    | 15.6    | 14.7 | 18.1 | 49.2        | 47.1        |
>
> SGSI consistently matches NRI’s training time (e.g., 20.4h vs. 20.1h on Spring Simulations and 47.1h vs. 47.2h for VN_NS_100), while methods such as iSIDG require ~1.5–2× longer. Yet recall that SGSI achieves 79.7% AUROC on Springs, which is higher than NRI. Overall, these results confirm that SGSI’s partial-knowledge gating imposes minimal additional cost while providing meaningful accuracy gains in structural inference.
>
> Thank you again for your helpful comments, and we hope these newly added experiments bolster confidence in SGSI’s robustness, scalability, and efficiency.

---

### Official Review · Reviewer_vBqs · 2025-03-13

**Overall Recommendation:** 4

**Summary:**

The paper introduces SGSI, a framework for latent graph structure learning that integrates partial prior knowledge into a VAE. It employs a soft gating mechanism with learnable parameters to smoothly control edge activation and uses a cloning and clamping strategy to fix known-present and known-absent edges without disrupting gradient flow. By enforcing adaptive regularization for global sparsity and node-degree constraints, SGSI effectively separates known from uncertain edges, optimizing the trade-off between compression and prediction via an information bottleneck perspective. Empirical results on diverse datasets—including physical simulations, biological networks, and multi-agent systems—demonstrate up to a 9% AUROC improvement over existing methods.

**Claims And Evidence:**

The paper’s main claims are generally well supported by both theoretical insights and empirical results.

**Essential References Not Discussed:**

They include many references. It seems quite comprehensive in general.

**Experimental Designs Or Analyses:**

The experimental design is generally sound. The authors evaluate SGSI on several benchmark datasets—including Spring Simulations, NetSim, synthetic biological networks, and vascular networks—and compare its performance against multiple baselines using AUROC as a metric. They vary the fraction of known-present and known-absent edges, which effectively demonstrates how partial prior knowledge improves inference accuracy. In addition, the paper includes detailed ablation studies (e.g., omitting the KL regularization on known edges or skipping the cloning step) to isolate the impact of each component on training stability and performance.

**Methods And Evaluation Criteria:**

I think they make sense.

**Other Comments Or Suggestions:**

N/A

**Other Strengths And Weaknesses:**

Strengths include its originality in integrating soft gating, cloning/clamping, and adaptive regularization to merge domain knowledge with latent structure learning. The paper is also theoretically grounded through an Information Bottleneck perspective and validated by strong empirical results.

Weaknesses involve limited analysis on scalability to large or heterogeneous graphs, modest hyperparameter sensitivity studies, and reliance on the availability and accuracy of prior knowledge.

**Questions For Authors:**

1. How robust is SGSI when the prior knowledge is noisy or partially inaccurate? How this will affect the approach proposed?

2. Can you provide additional experiments or analysis on the scalability of SGSI to larger and more heterogeneous graphs? A positive response with empirical evidence or theoretical insights would strengthen the scalability claim and improve my evaluation.

**Relation To Broader Scientific Literature:**

It builds on VAE-based structural inference methods (e.g., NRI by Kipf et al., 2018; Alet et al., 2019; Chen et al., 2021) that aim to learn latent graph structures, but goes further by incorporating partial prior knowledge—something earlier approaches tend to overlook.

The use of soft gating to control edge activation relates to established techniques in neural network regularization and gating mechanisms, ensuring smooth gradient flow compared to naive overwriting methods used in some prior works.

**Theoretical Claims:**

There are theorem statements in the paper.

---

> ### Author Rebuttal · Authors · 2025-03-31
>
> We thank you for your positive assessment and valuable comments. In response, we clarify our approach as follows (tables can be found at https://anonymous.4open.science/r/SGSI-Rebuttal-1614/Tables.pdf .)
>
> **1. Hyperparameters**
>
> SGSI introduces hyperparameters, most notably, the KL weight $\beta$, the sparsity penalty$ \lambda_{\mathrm{sparsity}}$, and the node-degree penalty $\lambda_{\mathrm{deg}}$. These parameters are essential for balancing the trade-off between inference and the incorporation of prior knowledge. We conducted Bayesian optimization with systematic grid searches on a validation subset to identify a “reasonable zone” that yields both high AUROC and stable convergence. Our initial search considered:
>
> - $\beta \in \\{0.1, 0.5, 1.0, 1.3, 1.4, 2.0, \dots\\}$,
> - $\lambda_{\mathrm{sparsity}} \in \\{10^{-4}, 10^{-3}, 10^{-2}\\}$,
> - $\lambda_{\mathrm{deg}} \in \\{10^{-4}, 10^{-3}, 10^{-2}\\}$.
>
> Our domain heuristics further guide these choices:
>
> - When the underlying graph is known to be very sparse (e.g., physical systems with local interactions), a larger $\lambda_{\mathrm{sparsity}}$ is preferable.
> - When node-degree constraints are well-established (e.g., each node has exactly 2 neighbors), a higher $\lambda_{\mathrm{deg}}$ ensures the model respects these constraints.
> - Conversely, if the prior knowledge is only approximate, lower penalty weights prevent the model from being over-constrained.
>
> We showcase results of sensitivity study on the VN_SP_30 dataset, and the results can be found in the Table 1 in the attached link. The results demonstrate that the optimal performance is achieved with $\beta=1.3, \lambda_{\mathrm{sparsity}}=0.004$, and $\lambda_{\mathrm{deg}}=0.01$, yielding an AUROC of 92.76%. Variations from these values result in a modest decrease in performance, which validates that while SGSI is indeed sensitive to these hyperparameters, it operates robustly within a reasonable range.
>
> **2. Robustness to Inaccurate Prior Knowledge**
>
> SGSI leverages soft penalties and cloned gating to integrate partial knowledge while preserving flexibility. Because these constraints are applied softly, via mild penalties and by clamping only a cloned copy of the gating vector, the model can deviate from incorrect priors when the data strongly contradicts them. In SGSI, the soft gating mask is applied after the Node-to-Edge operation (but not after Edge-to-Node), which helps preserve residual connectivity and prevents over-reliance on potentially flawed edges.
>
> To further validate this, we conducted “noisy prior” experiments on VN_SP_100 and VN_NS_100, where we randomly flipped 20% or 50% of the known-present/absent edges (with the overall prior knowledge set to 30%) and introduced 10% or 20% errors in global sparsity and degree constraints. Table 2 in the attached link summarizes our preliminary results.
>
> These results indicate that even with moderate noise, SGSI remains significantly more accurate than a no-knowledge baseline with at least 1~2% margin. Although performance gains naturally diminish as noise increases, the model robustly leverages available prior knowledge without catastrophic failure.
>
> **3. Scalability to Larger, Heterogeneous Graphs**
>
> We appreciate your interest in extending SGSI to larger or heterogeneous graphs. SGSI is designed to scale via mini-batching and subgraph sampling, approaches similar to GraphSAGE, which allow SGSI’s gating parameters to be computed or stored per subgraph rather than across the full $N \times N$ edge space. For extremely large graphs, we can either (i) restrict gating to local neighborhoods, or (ii) compute gating logits on the fly from node embeddings, thereby avoiding an $\mathcal{O}(N^2)$ parameter explosion.
>
> In this paper, we demonstrate scalability on the PEMS datasets (approximately 300 nodes) by leveraging multi-GPU mini-batching (in  Appendix E.2). To further validate SGSI on larger graphs, we generated a toy dataset with 5,000 nodes using Springs Simulations. Table 3 in the attached link shows the $\Delta$AUROC values under various levels of prior knowledge. These results indicate that SGSI, when using mini-batching, remains effective even at the 5k-node scale with prior knowledge, and with meaningful improvements in $\Delta$AUROC from 0.56 to 7.68.
>
> SGSI’s flexible architecture inherently supports heterogeneous graphs. By adopting per-relation gating parameters $\theta_{e,r}$ and applying the clone-and-clamp strategy to each edge type, SGSI can differentiate among various relations and node types while maintaining stable gradient flow and effective KL skipping. Although our current domain primarily involves homogeneous datasets, we recognize SGSI’s potential in applications such as multi-modal social networks, biomedical systems, and multi-layer transportation networks, and plan to explore these in future work.
>
> Thank you again for your positive feedback. We look forward to refining our manuscript with these additional experiments.

---

> > ### Comment · Reviewer_vBqs · 2025-04-04
> >
> > Thank you to the authors for their insightful rebuttal. Although I am not very familiar with the topic, I can see that the authors have made a sincere effort to address all of my concerns, and their responses are reasonable. Given that my initial score was already a 4, which is quite high, I would prefer to keep my score unchanged. Thanks for your response!

---

> > > ### Author Response · Authors · 2025-04-05
> > >
> > > We sincerely thank you for your thoughtful evaluation and positive feedback on our rebuttal. Your comments were extremely helpful in improving our manuscript. We completely understand and respect your decision to keep your score unchanged given your initial strong support.
> > >
> > > Thanks again for your valuable insights and the encouraging review!

---

### Decision · Program_Chairs · 2025-05-01

**Decision:**

Accept (poster)

**Comment:**

This paper presents Soft-Gated Structural Inference (SGSI), a framework for incorporating partial prior knowledge into VAE-based structural inference. The proposed method uses a combination of soft gating, clone-and-clamp mechanisms, and adaptive regularization to handle known edge constraints, sparsity, and degree priors in a stable and interpretable way.

The reviewers generally appreciated the novelty and empirical strength of the method. Multiple reviewers praised the paper for integrating domain knowledge into structural inference without destabilizing learning, noting that SGSI achieves notable gains in AUROC and provides interpretability and robustness. Concerns were raised regarding the practical availability of prior knowledge, the technical novelty of certain components (e.g., clone-and-clamp), and the lack of downstream task evaluation, which the authors addressed with new experiments and real-world data. These clarifications were well-received by the reviewers.

Overall, this submission is well-positioned for acceptance, as it provides a practical, extensible solution to an important problem in structure learning, with strong experimental validation and thoughtful design choices. Therefore, I recommend acceptance.